# Balancing the Growth Performance and Nutritional Value of Edible Farm-Raised Sago Palm Weevil (*Rhynchophorus ferregineus*) Larvae by Feeding Various Plant Supplemented-Sago Palm Trunk Diets

**DOI:** 10.3390/foods12183474

**Published:** 2023-09-19

**Authors:** Athakorn Promwee, Khanittha Chinarak, Worawan Panpipat, Atikorn Panya, Natthaporn Phonsatta, Matija Harcet, Manat Chaijan

**Affiliations:** 1Food Technology and Innovation Research Center of Excellence, School of Agricultural Technology and Food Industry, Walailak University, Nakhon Si Thammarat 80160, Thailand; athakorn.pr@wu.ac.th (A.P.); khanittha.ch@mail.wu.ac.th (K.C.); cmanat@wu.ac.th (M.C.); 2Food Biotechnology Research Team, Functional Ingredients and Food Innovation Research Group, National Center for Genetic Engineering and Biotechnology (BIOTEC), 113 Thailand Science Park, Phaholyothin Rd., Khlong Nueng, Khlong Luang, Pathum Thani 12120, Thailand; atikorn.pan@biotec.or.th (A.P.); nat-thaporn.pho@biotec.or.th (N.P.); 3Division of Molecular Biology, Ruđer Bošković Institute, Bijenička 54, 10000 Zagreb, Croatia; matija.harcet@irb.hr

**Keywords:** edible insect, palm weevil larvae, growth performance, plant-based ingredients, nutritional value, lipid metabolism

## Abstract

Herein, the effect of supplementing ground sago palm trunk (GSPT) with varying concentrations of plant-based ingredients (PIs), including rice bran (RB), soybean meal (SM), and perilla seed (PS), on the nutritional profile of sago palm weevil larvae (SPWL) was investigated. Increased PS intake induced an increase in α-linolenic acid level and a reduction in the n-6/n-3 ratio in SPWL (*p* < 0.05). The presence of fatty acids in SPWL was determined predominantly by the fatty acid profile in the feed. The activities of Δ5 + Δ6 desaturases and thioesterase were not different among SPWL fed different diets (*p* < 0.05); however, PI intake resulted in low suppression of *fads*2 gene expression. RB, SM, and PS at the appropriate concentrations of 17.5%, 8.8%, and 7.0% in GSPT (F3 diet), respectively, boosted both protein quantity and quality of SPWL, as indicated by higher levels of essential amino acids, particularly lysine, than the FAO protein reference. Therefore, incorporating PIs into a regular diet is a viable method for enhancing the nutritional value and sustainability of farm-raised SPWL as a potential alternative source of high-quality lipid and protein.

## 1. Introduction

Edible insects are currently identified as a viable source of vital nutrients for dietary purposes, with the hope of helping support food security and sustainability in the future. They also have the ability to recycle waste materials and agricultural byproducts in order to provide high-quality nutrition for humans and various domesticated animals [1]. According to life cycle analysis, producing insects uses less land, feed, water, and other bioresources than traditional livestock while emitting fewer greenhouse gases and having a smaller impact on climate change [2]. Furthermore, edible insects could help to improve global food security by preventing famine in countries with limited economic resources [3]. Currently, China has taxonomically categorized more than 300 species of edible insects [4]. These insect species belong to the orders Blattaria, Coleoptera, Diptera, Ephemeroptera, Hemiptera, Hymenoptera, Isoptera, Megaloptera, Lepidoptera, Orthoptera, and Odonata, with Lepidoptera, Coleoptera, and Hymenoptera accounting for the majority of these species [4]. The most prevalent insects are locusts, cicada, diving beetles, mealworms, bamboo worms, silkworms, tussah, Italian and Oriental honeybees, wasps, and black ants [4,5]. Insect food safety studies show that these insects are safe to consume [6]. However, care must be taken to prevent contamination from potential insect hazards, such as pathogens, toxins, pesticide residues, and heavy metals, during rearing and harvesting [6].

The sago palm weevil (*Rhynchophorus ferrugineus*) belongs to the order Coleoptera [7]. The order Coleoptera is categorized as safe for consumption according to an evaluation of the toxicological features of edible insects [6]. According to Thai traditional knowledge and practices involving the usage of edible insects, including toxicological facts that are retained from memory, sago palm weevil larvae (SPWL) are safe for human consumption. Because of their excellent nutritional content and distinct taste and flavor, SPWL are widely consumed in numerous regions of the world, particularly in Asia and Africa [8,9,10,11]. Trade in edible insects, specifically SPWL, holds immense promise because some countries, like Thailand, even grow SPWL on a commercial scale for the food industry [12]. Consumer acceptance of products derived from SPWL and African palm weevil larvae has been studied [13,14]. The sensory features of foods like seasoning powder based on SPWL were identical to those of commercial products made from pork and chicken [13]. Consumer approval of enriched supplemented foods increased dramatically when fermented sorghum and maize flours were fortified with African palm weevil larvae powder [14].

SPWL grow naturally in sago palms (*Metroxylon sagu*, *M. rumphii*) [8]. However, unsustainable harvesting of wild larvae has resulted in a decline in their population and increased interest in developing sustainable production methods. Controlled breeding and farm-rearing of SPWL are the solutions to increasing production quantity and quality on a commercial scale. To increase the growth rate of SPWL, a commercial raising method was recently tested by feeding SPWL ground sago palm trunk (GSPT) plus commercial pig feed (PF) in a plastic bowl. The resulting SPWL grew well, but they were high in saturated fatty acids (SFAs) and cholesterol while low in polyunsaturated fatty acids (PUFAs) and essential amino acids, resulting in an imbalanced nutritional value [8]. As a consequence, feed composition influences the nutritional value and growth performance of SPWL. The use of plant-based ingredients (PIs), particularly perilla seed (PS), rice bran (RB), and soybean meal (SM), may satisfy the nutritional needs of consumers for SPWL, such as high protein, rich in essential amino acids/fatty acids, balance in n3/n6 PUFAs, low in cholesterol, and rich in minerals. Our recent study on the influence of dietary supplements on the growth rate and nutritional status of SPWL found that feeding SM mixed with the basal diet could improve the essential amino acid content of SPWL while having no effect on essential fatty acids [15]. Mealworms (*Tenebrio molitor*) raised on a wheat bran-based diet developed faster and had a higher protein content [1]. Dietary inclusion of PS in GSPT increased levels of n-3 PUFAs, essential amino acids, and protein content of SPWL while decreasing the n3/n6 ratio [15]. The PS-formulated diet, by contrast, had a negative impact on SPWL growth and survival, and SPWL had a lower mineral content during long-term feeding. Western corn rootworms (*Diabrotica virgifera* LeConte) reared on mixed diets (wheat germ, corn root powder casein, and linseed oil) showed increased weight gain, survival rate, and molting rate [16]. All of the previous studies demonstrated that each ingredient had negative or positive effects on the growth performance and nutritional value of edible insects, implying that a mixed diet could be a solution to maximize both nutrient and growth prospects [1,8,15,16].

This research aims to fine-tune the proper feed composition using agricultural PIs (SM, RB, and PS) to balance the growth performance and nutritional value of SPWL while avoiding any negative effects. This was predicated on the hypothesis that dietary intake significantly affects how well SPWL grow and their nutritional value. The nutritional composition of the final SPWL and nutritional requirements were used as the selection criteria for each plant-based ingredient, each of which had its own predominant nutrients. The findings may aid in the development of more sustainable and nutritious diets for SPWL, as well as conservation of this vital food resource.

## 2. Materials and Methods

### 2.1. Chemicals

The chemicals and solvents used in this investigation were all of gas chromatography (GC) and analytical grade, purchased from Sigma-Aldrich (St. Louis, MO, USA).

### 2.2. Experimental Diets 

The optimization of supplementing PIs in the basal ground sago palm trunk (GSPT; 66.7%) with varying concentrations of soybean meal (SM; 4.4–11.1%), rice bran (RB; 8.9–22.2%), and perilla seed (PS; 0–20%) was investigated (Table 1) in comparison to the basal diet alone (Control; C) and the basal diet supplemented with pig feed (PF). Two types of feed are often utilized on farms for SPWL, depending on the growing region: (1) GSPT alone, which represents natural raising, and (2) GSPT enriched with PF, which is frequently added to the basal diet of many farm-raised SPWL to boost the growth rate. One of the factors that affects their utilization is religious conviction. For instance, farmers who are not restricted by their religious beliefs supplement their feeds with PF, whereas Muslim farmers are prohibited from doing so. Thus, both diets were included in this study. The nutritional content, growth performance, lipid metabolism indicators, and gene expression of Δ6 desaturase (*fads*2) in SPWL were all thoroughly examined. To prepare the supplemental diets, GSPT was mixed with each supplement in the specific content shown in Table 1 until the feed was homogeneous. PS was purchased from Bankongloi, Chiangmai, Thailand, and PF, RB, and SM were procured from Thasala market, Nakhon Si Thammarat, Thailand. The costs of GSPT, PF, RB, SM, and PS were approximately 0.13, 0.22, 0.26, 0.32, and 1.70 USD/kg [15].

According to Chinarak et al. [15], SPWL were raised at room temperature (28–30 °C) with 70–80% relative humidity. Four adult weevil pairs were grown in a spherical plastic container containing 3 kg of feed and 1 L of water. As shown in Table 1, different diet formulations containing 66.7% GSPT mixed with varying concentrations of SM (4.4–11.1%), RB (8.9–22.2%) RB, and PS (0–20%) were used in rearing SPWL in separate containers. The larvae were developed after 20 days, and then the rearing period was extended to 40 days. The number of larvae in each group was set at 120 larvae per container. On Days 20 and 30, newly made diet (1.5 kg each time) was introduced in order to prevent a feed shortfall. The rearing stage was divided into three batches. Growth performance was measured after 20 and 40 days of rearing. To determine the nutritional value, 40-day-old larvae were rinsed with tap water, blanched with boiling water (5 min), cooled in ice water (10 min), freeze-dried using a freeze-dryer (FTS systems Inc., Stone Ridge, NY, USA), and stored in a plastic box at −20 °C until use.

### 2.3. Growth Performance

Biometric characteristics were determined using 20 larvae from each group. After 20 and 40 days, the larvae were measured for total length (cm) and body weight (g) using a vernier caliper (Mitutoyo 530–312, Tokyo, Japan) and a digital balance (Sartorius AG, Goettingen, Germany), respectively. The biometric parameters of SPWL were computed using the following formulae:(1)$$Dry\;matter\;content\;\left(\%\right)=(dry\;weight/live\;weight)\times 100$$
(2)$$Condition\;factor=(body\;mass\times 100)/{body\;length}^{3}$$
(3)$$Survival\;\left(\%\right)=(number\;of\;final\;larvae/number\;of\;initial\;larvae)\times 100$$
(4)Growth rate (g/day)=(ln final body weight−ln initial body weight)/day

### 2.4. Proximate Composition

The proximate composition (moisture, crude fat, crude protein, and ash) of the freeze-dried SPWL was determined [17]. The carbohydrate content was estimated by subtracting the sum of the components from 100. A conversion factor of 5.6 was chosen to prevent an overestimation of the protein level because of the existence of non-protein nitrogen in SPWL [8,15]. 

### 2.5. Mineral Composition

Minerals were identified using the AOAC method [17] with an inductively coupled plasma optical emission spectrophotometer (PerkinElmer, Model 4300DV, Norwalk, CT, USA). Sodium (Na), magnesium (Mg), potassium (K), phosphorus (P), manganese (Mn), calcium (Ca), iron (Fe), zinc (Zn), and copper (Cu) were reported.

### 2.6. Fatty Acid Composition 

According to Chinarak et al. [8], the FA profile of oil samples in the form of fatty acid methyl esters (FAMEs) was analyzed using a GC/quadrupole time-of-flight (GC/Q-TOF) mass spectrometer (GC 7890B/MSD 7250, Agilent Technologies, Santa Clara, CA, USA).

The n-3/n-6 ratio, sum of the SFAs, unsaturated fatty acids (UFAs), monounsaturated fatty acids (MUFAs), PUFAs, and ratios of PUFAs to SFAs were calculated. The index of atherogenicity (IA) and index of thrombogenicity (IT) were calculated using the following formulae [15].
(5)$$IA=\frac{C12\text{:}\;0+4\times C14\text{:}\;0+C16\text{:}\;0}{\Sigma MUFA+\Sigma PUFA\;\omega -6+\Sigma PUFA\;\omega -3}$$
(6)$$IT=\frac{C14\text{:}\;0+C16\text{:}\;0+C18\text{:}\;0}{0.5\Sigma MUFA+0.5\Sigma PUFA\omega -6+3\Sigma PUFA\;\omega -3+\frac{PUFA\;\omega -3}{PUFA\;\omega -6}}$$

### 2.7. Amino Acid Composition

The amino acid composition of SPWL was analyzed according to Chinarak et al. [15] using a Shimadzu GCMS-TQ8050 NX (Kyoto, Japan). The concentration of amino acids in each dry sample was expressed in milligrams per gram. The essential amino acid index (EAAI) and biological value (BV) were calculated using the formulae proposed by Kulma et al. [18]:(7)$$EAAI=\sqrt[{9}]{\frac{g\;of\;lysine\;in\;100\;g\;of\;analysis\;protein\times 100}{g\;of\;lysine\;in\;100\;g\;of\;reference\;protein}\times (etc.\;for\;other\;8\;EAA)}$$
(8)BV=1.09×EAAI−11.7

### 2.8. Cholesterol Content

The cholesterol levels of the diets and the resulting SPWL were determined using a gas chromatography-triple quadrupole mass spectrometer (GC/QQQ, GC 7890B/MSD 7000D, Agilent Technologies, Santa Clara, CA, USA) [19].

### 2.9. Fatty Acid Metabolism Indices 

Desaturase indices were estimated by dividing the proportion of the product by the percentage of the precursor [20], and they were used to assess the activity of desaturating enzymes in converting SFAs to MUFAs:(9)Δ9−desaturase (18) index: ∆9−DI (18) =100 [C18: 1/(C18: 1+C18: 0)]
(10)Δ9−desaturase (16) index: ∆9−DI (16) =100 [C16: 1/(C16: 1+C16: 0)
(11)Total ∆9−DI=100 [(C16: 1+C18: 1)/(C16: 1+C16: 0+C18: 1+C18: 0)]

The activities of the Δ5 + Δ6 desaturase index (DI) and thioesterase and elongase indices were calculated as follows [20]:Δ5 + Δ6 − DI = 100 [C20: 2ω−6 + C20: 4ω−6 + EPA + C22: 5ω−3 + DHA/C18: 2ω−6 (LA) + ALA + C20: 2ω−6 + C20: 4ω−6 + EPA + C22: 5ω−3 + DHA](12)
(13)Elongase index (EI) =C18: 0/C16: 0
(14)Thioesterase index (TI) =C16: 0/C14: 0

### 2.10. Fads2 Gene Expression in SPWL

The expression of Δ6 desaturase (*fads*2) in SPWL was determined using quantitative real-time PCR (qRT-PCR), as described by Chinarak et al. [21]. Degenerate primers for *fads*2 were created using known desaturase sequences from *Danio rerio*, *Mus musculus*, *Rattus norvegicus*, *Argyrosomus regius*, and *Nibea mitsukurii* (GenBank accession numbers BC049438.1, KC261978.1, BC057189.1, and GQ996729.1, respectively). The relative gene expression in SPWL was measured using β-actin (Forward 5′-GATTCTGGAGATGGT, Reverse 5′-TCTGGGCAACGGAAC) and *fads*2 (Forward 5′-TGAACCAGTCRTT GAAG, Reverse 5′-GCTG-GATGGYTRCARC) primers. Following the manufacturer’s instructions, total RNA was extracted from SPWL using TRIZOL reagent (Invitrogen, Carlsbad, CA, USA). RNA concentration and purity were determined at 260 and 280 nm, respectively. Agarose gel electrophoresis (1% *w*/*v*) was used to assess the integrity of the RNA. RNA was reverse transcribed into first-strand cDNA using reverse transcriptase (iScriptTM Select cDNA Synthesis Kit, Bio-Rad, Hercules, CA, USA). Using qRT-PCR, the *fads*2 mRNA level in SPWL was determined. For qRT-PCR, cDNA (template, 1 μL), forward and reverse primers (10 pmol/μL each, 0.25 μL), 5 × Hot FIREPol Evareen qPCR Mix Plus (ROX, 2 μL), and PCR grade water (6.5 μL) were combined in a reaction mixture (10 μL). The denaturation temperature cycle was performed at 95 °C/12 min, 95 °C/30 s, 40 cycles of 55–60 °C/ 30 s, and 72 °C/30 s. The relative expression of the *fads*2 gene in relation to that of β-actin (the reference gene) was computed using the comparative threshold cycle (Ct) method.

### 2.11. Statistical Analysis 

Statistical Package for the Social Sciences (SPSS) 24 for Windows (SPSS Inc., Chicago, IL, USA) was employed for the statistical evaluation. Each sample was examined 3 times. For statistical purposes, one-way ANOVA was utilized. Duncan’s multiple range test was performed to find significant differences (*p* < 0.05) among samples while comparing means.

## 3. Results and Discussion

### 3.1. Nutritional and Fatty Acid Compositions of Experimental Diets

SPWL grow naturally in sago palm trunks, and GSPT is a vital ingredient in the production of farm-raised SPWL. GSPT is typically mixed with commercial animal feed, particularly pig feed, to improve the growth rate of commercially farmed SPWL in Thailand [8]. There are three major drawbacks to using commercial pig feed-formulated diets: (1) the uncontrollable production costs due to pig feed price fluctuations, (2) the lack of essential fatty acids and essential amino acids, and (3) the limited acceptance by Muslim people, resulting in a narrow market. Alternative PI-formulated diets derived from the agricultural sector could provide a viable option for producing sustainable farm-raised SPWL. 

In this study, the feed composition was optimized using three PIs to improve the nutritional value of SPWL while maintaining a proper growth rate. Table 2 shows the proximate and fatty acid compositions of the feed mixtures. The crude protein, carbohydrate, fat, and ash contents varied according to the type of plant-based ingredient employed (Table 1). All tested diets had moisture contents ranging from 53.8% to 68.8%, with the baseline reference diet having the greatest level of moisture (*p* < 0.05). The addition of dried PIs or PF in various proportions resulted in decreases in the moisture contents of the corresponding diet formulations (Table 2). Carbohydrate was the most abundant component (62.2–94.1%) found in all feed formulations due to the presence of starch in GSPT. All plant ingredient-based formulated diets (F1–F5) had higher protein contents (11.3–13.4%), lipid contents (6.2–21%), and ash contents (4.3–8.1%) than the control diet (*p* < 0.05). The F1 formulation had the highest crude protein content (13.4%), which corresponded to the highest amounts of added RB (22.2%) and SM (11.1%) in this formulation (*p* < 0.05). However, the presence of protein in the tested diets was not related to the concentrations of added RB and SM in a linear fashion. Because all of the plant ingredient-based tested diets contained four components (GSPT, RB, SM, and PS), the level of naturally occurring protein in each component resulted in the net protein content of the final formulation. It should be noted that increasing the PS concentration led to a gradual rise in the lipid content of the resulting diet (Table 2). The F5 formulation showed the greatest lipid content of 21% (*p* < 0.05), which corresponded to the highest amount of PS (20%) present. As a result, the presence of PS was an important factor in controlling the lipid content of the tested diet. The PF-formulated diet contained the most ash (*p* < 0.05), which could be attributed to the added minerals in pig feed. Therefore, the variation in the proximate composition of the experimental diets was strongly influenced by the type and concentration of the particular ingredients.

The fatty acid profiles of all diets are shown in Table 2. The major fatty acids in the control GSPT diet were palmitic acid (C16:0), stearic acid (C18:0), and oleic acid (C18:1), accounting for 57.5 g/100 g of lipid. Linoleic acid (C18:2), an essential omega-6 fatty acid, was found in all PI-based diets, implying that RB, SM, and PS are sources of linoleic acid. The PF diet, on the other hand, provided the highest linoleic acid content, while the control basal diet had the lowest linoleic acid content (*p* < 0.05). Interestingly, there was a significant increase in the α-linolenic acid (C18:3) content as the proportion of PS supplementation increased, which was 9- to 27-fold greater than that in the control diet (*p* < 0.05). This scenario can be explained by the high concentration of naturally occurring α-linolenic acid in PS [15]. An increase in the α-linolenic acid content led to a rise in the overall n-3 PUFA level and a reduction in the n-6/n-3 ratio (Table 2). The same trend was documented by Deng et al. [22], who found a high α-linolenic acid content in PS-mixed lamb feed. It should be noted that the addition of PIs reduced the total SFA content while increasing the total PUFA and n-3 PUFA contents (Table 2).

### 3.2. Growth Performance of SPWL

The ability of SPWL to grow when cultivated on farms is a crucial component of their commercial competitive advantage. Table 3 depicts the appearances and growth rates of SPWL fed different dietary regimens. The growth parameters of SPWL raised on different PI-based diets differed significantly from those raised on the PF and basal (control) diets, particularly their survival rates, condition factors, and live weights after 40 days of rearing. The survival rates did not differ significantly among SPWL fed the PF, F1, F2, and F3 diets and the control diet (*p* > 0.05), which were all around 90% (Table 3). SPWL raised on the F4 and F5 diets had lower survival rates (90%), despite the fact that both formulations contained low concentrations of RB and SM with a high PS content (Table 1). This was consistent with what we had previously reported [15] regarding the reduction in SPWL survival rate by feeding a high PS-based diet. Owing to the existence of antinutritive elements, notably phytic acid, and the high fat content of PS, the increased PS content of the feed may have decreased larval survival [15]. Perilla ketone, an antinutritive terpenoid that significantly affects how well the digestive tracts of animals function, is also a component of PS [23]. Pulmonary edema and signs of perilla mint toxicosis may be induced by increased intake of perilla ketone and other antinutritive substances in the body. As a result, nutrient imbalance and the presence of antinutritive agents in the F4 and F5 diets may have interfered with larval metabolism, resulting in lower survival rates (Table 3). According to Starčević et al. [24], the addition of linseed oil (3–5%) and pumpkin seed oil (3–5%) to cricket diets showed a detrimental effect on the survival rate, which was reduced from 38% to 22% and 45% to 35%, respectively.

SPWL fed the F2 (20.2% RB, 10.1% SM, and 3.0% PS) and F5 (8.9% RB, 4.4% SM, and 20.0% PS) diets exhibited the greatest live and dry weights (*p* < 0.05). The specific amounts of three PIs in both formulations altered the live and dry weights, with the balanced nutrient intake in the F2 diet promoting the larval growth rate and the high fat intake in the F5 diet resulting in high live and dry weights. To determine the proper growth performance, all survival rates and larval weights should be examined. SPWL fed the F5 diet exhibited the greatest dry matter content (*p* < 0.05). There were no significant differences (*p* > 0.05) in the condition factors of SPWL fed the F1, F2, F3, and F5 diets, which were markedly higher than those of SPWL fed the PF and control diets (*p* < 0.05). All SPWL fed PI-supplemented diets exhibited slower growth rates than those fed the PF and control diets (Table 3), with the PF diet leading to the fastest growth rate (*p* < 0.05). Surprisingly, these growth rates had no relationship with live weight because the PI-treated groups grew faster in the first 20 days and then slower in the last 20 days. However, when fed diets supplemented with proper ratios of three PIs, the survival rates of SPWL were approximately as high. Thus, incorporating appropriate amounts of RB, SM, and PS into regular GSPT can improve SPWL growth performance to levels comparable to or greater than those achieved by feeding an animal feed-formulated diet.

### 3.3. Basic Nutritional Composition of SPWL

Table 4 shows the proximate compositions of SPWL fed various experimental diets. The variation in basic nutritional value of SPWL was governed by the type and concentration of the individual ingredients fortified in the diet formulations. All SPWL had moisture contents ranging from 57.5% to 72.6% (Table 4), with SPWL fed the PF, F3, and control diets showing the greatest moisture levels (*p* < 0.05). It should be noted that the lowest moisture content (57.5%) was found in SPWL fed the F5 diet (*p* < 0.05), the formulation with the highest PS and lipid contents, but the decreased moisture level of SPWL was not proportional to the increased PS content of the tested diets (Table 4). SWPL fed the F3 diet (containing 17.5% RB, 8.8% SM, and 7.0% PS) had the highest crude protein content (26.8%) (*p* < 0.05), followed by those fed the F1 ≈ F2 ≈ F4, F5 ≈ control, and PF diets, respectively. Interestingly, the F3 formulation did not contain any predominant PI as a protein source, suggesting that when used in appropriate amounts, all three PIs had a synergistic effect on protein accumulation in SPWL. Since the lower limit for labeling a “high protein food” product is 10 g/100 g edible portion [25], SPWL reared on any PI-based diet with approximately 20 g protein/100 g insects can be considered a high-protein food source. In general, insect proteins are regarded to be identical to the proteins present in human bodies, while having a high protein level similar to that of meat and fish, making them simpler to absorb than plant-based proteins [26]. The protein contents of SPWL fed PI-fortified diets were greater than those of *Phassus triangularis* (15 g/100 g), *Tribolium castaneum* (17.0 g/100 g), *Imbrasia truncate* (19.1 g/100 g), and *Imbrasia epimethea* (20.1 g/100 g) [3,25,27]. Nevertheless, the protein contents were less than those of *Zophobas morio* (43.13 g/100 g), *Gryllus assimilis* (65.52 g/100 g), *Acheta domesticus* (64.38 g/100 g), *Hermetia illucens* (33.91 g/100 g), and *Brachygastra mellifica* (38.24 g/100 g) [28,29].

The lipid contents of SPWL raised on various diet formulations ranged from 33.0 to 58.1 g/100 g (Table 4). SPWL fed the PF, F2 (20.2% RB, 10.1% SM, and 3.0% PS), and F4 (12.2% RB, 6.1% SM, and 15.0%) diets showed the greatest fat contents (*p* < 0.05). The addition of supplemental ingredients, both plant- and animal-based, to GSPT increased the lipid contents of the corresponding SPWL by 1.33–1.76-fold when compared to those fed the basal diet (Table 4). Surprisingly, the lipid content of the resulting SPWL did not correlate linearly with the fat content of the tested diets, with the F5 diet (highest lipid content, 21.0%) inducing the lowest lipid accumulation in SPWL (44.0% lipid content, *p* < 0.05). The lipid content of the feed could not be stored entirely in the insect’s body, and the ingredient composition of the feed may play an important role in controlling lipid accumulation in SPWL. The lipid contents of all SPWL fed PI-based diets were higher than those of *Allomyrina dichotoma* (20.24 g/100 g), *Protaetia brevitarsis* (15.36 g/100 g), *Tenebrio molitor* (34.54 g/100 g), *Teleogryllus emma* (25.14 g/100 g), and *Gryllus bimaculatus* (11.88 g/100 g) [30].

The ash contents of all SPWL fed PI-based and PF diets were significantly lower than those of SPWL fed the basal diet (*p* < 0.05), but there was not a noticeable distinction between SPWL fed the PF, F1, F2, and F3 diets (*p* < 0.05). Owing to the presence of the anti-mineral agent, phytic acid, the level of ash of SPWL fed PI-formulated diets dropped as the proportion of PS rose in the studied diets [15]. The tested diet, which exhibited the greatest content of ash (PF diet, Table 2), had no relation to the ash content of the corresponding SPWL, which had a moderate ash content (3.0 g/100 g, Table 4). This circumstance may be explained by mineral homeostasis in this insect, in which lower mineral absorption occurred when feeding a high mineral-containing diet. All SPWL fed PI-supplemented diets had lower carbohydrate contents (15.1–32.4 g/100 g) than SPWL fed the regular diet (40.6 g/100 g), with SPWL fed the control diet showing the greatest carbohydrate content (*p* < 0.05).

The high carbohydrate content of the GSPT in the control diet (Table 2) was responsible for the presence of the greatest amount of carbohydrate in the corresponding SPWL (Table 4). It should be noted that while all PI-based and PF diets had the same GSPT concentration (66.7%), the accumulated carbohydrate in the resulting SPWL varied according to the type and concentration of the supplemental ingredients. SPWL fed the F2 (20.2% RB, 10.1% SM, and 3.0% PS) and F3 (17.5% RB, 8.8% SM, and 7.0% PS) diets had the lowest carbohydrate contents (*p* < 0.05), which was unrelated to the presence of carbohydrates in the tested diets (Table 2). Surprisingly, the second highest carbohydrate content (32.4 g/100 g) was found in SPWL raised on the F5 diet (8.9% RB, 4.4% SM, and 20.0% PS), which had the lowest carbohydrate and highest lipid contents (Table 2). This implied that a high lipid intake could be converted to stored carbohydrate during insect metabolism. Although the influence of PI type and concentration on the proximate composition of SPWL was unclear, both parameters appeared to exert influences in a nonlinear fashion. From the results, the type and concentration of each ingredient in the feed had a significant impact on the basic nutritional components in the resulting SPWL.

### 3.4. Mineral Composition of SPWL

Edible insects are of great interest as a source of minerals such as Fe, Zn, K, Na, Ca, P, Mg, Mn, and Cu [31]. SPWL had higher levels of macro-elements, specifically K, P, Mg, Ca, and Na (Table 4). The most prevalent mineral in all SPWL was K (5652–9417 mg/kg), which was present at higher or comparable levels to traditional muscle foods such as pork (5043 mg/kg) and chicken (5557 mg/kg) [30]. The highest K concentration was found in SPWL fed the control diet, followed by those fed the F3, F1, F2 ≈ PF, F4, and F5 diets (*p* < 0.05). The amount of RB in PI-mixed diets appeared to affect the K content of SPWL. The high level of K in GSPT was possibly responsible for the highest concentration of this element in the corresponding SPWL. SPWL had a higher K level than *Bombay locust* (3498 mg/kg), scarab beetle (5208 mg/kg), house cricket (4577 mg/kg), mulberry silkworm (4929 mg/kg), *Zonocerus variegatus* (2551 mg/kg), *Macrotermes bellicosus* (2738 mg/kg), and *Cirina forda* (2373 mg/kg) [32,33].

P is readily available in insects, as opposed to P derived from plants [30]. The P concentration varied from 2869 to 4003 mg/kg (Table 4), with SPWL fed the F3 diet showing the greatest P concentration (*p* < 0.05). The lowest P level was observed in SPWL fed the F5 diet, which was statistically identical to that in SPWL fed the PF diet (*p* > 0.05), possibly due to the low ash content of this formulation and the presence of a high phytic acid level. There was no trend in changing P content among all SPWL fed PI-based diets (Table 4), implying that the specific ratios of each PI influenced the presence of P in SPWL. Furthermore, no linear interrelationship was noted between the P level in SPWL (Table 4) and the ash content of the diet (Table 2). The P contents of all SPWL were greater than those of *Oryctes rhinoceros* (388 g/kg) and *Zonocerus variegatus* (1817 mg/kg) [34], which met the daily intake recommendation for healthy individuals (700–4000 mg) by the World Health Organization (WHO) [35].

Mg is required for a variety of vital activities in chronic kidney disease (CKD), including neuromuscular, enzymatic reactions, neurotransmitter release, and mineral and bone disease metabolism [36]. SPWL reared on various diets had Mg contents ranging from 1240 to 2082 mg/kg (Table 3). SPWL fed the F3 diet (17.5% RB, 8.8% SM, and 7.0% PS) showed the greatest Mg level (*p* < 0.05), surpassing that of *Rhynchophorus phoenicis* (336–1318 mg/kg), *Bombxy mori* (2070 mg/kg), and *Ruspolia differens* (331 mg/kg) [37]. Except for SPWL raised on the F5 diet, all SPWL fed PI-based diets had higher levels of Mg than those fed the PF diet (*p* < 0.05), demonstrating that fortification with three PIs in a specific ratio resulted in higher Mg levels in this insect. The Mg content of SPWL (Table 4) was not dependent on the ash content of the diets (Table 2), implying that element absorption in this insect plays a key role in the net storage of individual minerals.

Na is required for proper cellular homeostasis, fluid and electrolyte balance, blood pressure, the excitability of nerve cells and muscle, and the movement of nutrients and substrates across plasma membranes [38]. However, daily Na intake is limited to 1500 mg/day for people with hypertension and less than 2300 mg/day for healthy individuals [39]. The Na levels of SPWL ranged from 827 to 1374 mg/kg (Table 4), with SPWL fed the F3 and control diets providing the greatest Na contents (*p* < 0.05). These, however, had lower Na contents than *Oryctes rhinoceros* larvae (9315 mg/kg) [34]. There was also no relationship between the Na content of SPWL and the ash content of the corresponding diets (Table 2), suggesting that mineral metabolism controls the amount of mineral stored in this insect.

Ca is an essential mineral involved in neuromuscular function, enzyme-mediated functions, bone and teeth formation, and blood coagulation [40]. Overall, SPWL had Ca contents ranging from 431 to 731 mg/kg (Table 4), which were significantly higher than those of typical animal-based foods such as beef (187 mg/kg), chicken (323 mg/kg), and pork (379 mg/kg) [30]. SPWL fed the F3 diet showed the greatest Ca level, followed by those fed the control, F1, F2, F4, and F5 ≈ PF diets (*p* < 0.05). It should be noted that the increased Ca contents of SPWL fed PI-based diets was linear up to 15% PS (F1–F4), then abruptly dropped when PS was increased to 20% (F5). All SPWL had higher Ca contents than *Zophobas morio* (319 mg/kg) and *Gryllus assimilis* (453 mg/kg) [41].

The most common microelements found in insects are Zn, Mn, Fe, and Cu. SPWL are an excellent source of microminerals, with 78.1–131.6 mg/kg of Zn and 13.2–18.8 mg/kg of Fe (Table 4). SPWL fed the F3 diet had the highest Zn (131.6 mg/kg) and Fe (18.8 mg/kg) contents, which were higher than those of *Oryctes rhinoceros* with 6.5 mg/kg of Zn and 12 mg/kg of Fe [34]. *Cirina forda*, on the other hand, had higher Zn (138 mg/kg) and Fe (278 mg/kg) contents than the current SPWL [33]. Increasing the PS content from 0% to 7% resulted in proportional increases in both Zn and Fe contents in SWPL fed PI-based diets, followed by decreases in levels of both elements after adding 15–20% PS. The Cu contents of SPWL reared on various diets ranged from 10.0 to 25.9 mg/kg (Table 4), with the greatest concentration found in SPWL fed the PF diet, followed by those fed the F3, F4, F1, F5, control, and F2 diets (*p* < 0.05). Cu enrichment in SPWL fed the PF diet may have been linked to the presence of a high Cu content of PF. Cu deficiency in children and adults can typically result in slowed cardiovascular development, bone deformity, and long-term neurologic and immunologic problems [42]. SPWL’s Mn contents varied from 10.4 to 49.9 mg/kg (Table 4), with the greatest content observed in SPWL fed the control diet (*p* < 0.05). Lower Mn levels were noticeable in all SPWL fed the supplemented diets compared to those fed the control diet (*p* < 0.05). Mn is involved in bone formation, amino acid, carbohydrate, and cholesterol metabolism, as well as the catalytic activity of Mn metalloenzymes like glutamine synthetase, arginase, manganese superoxide dismutase, and phosphoenolpyruvate decarboxylase [43].

Overall, SPWL fed the F3 diet (66.7% GSPT, 17.5% RB, 8.8% SM, and 7.0% PS) had superior macro- and micromineral levels, which were comparable to or even greater than those fed the PF and control diets. As a result, SPWL are a potential alternative source of essential elements, especially when fed a diet based on a specific ratio of PIs. This study demonstrates SPWL as a potential mineral source for human and animal consumption. Since humans require at least 22 minerals involved in metabolic processes, the presence of mineral diversity in SPWL results in a desired quality for future food consumption.

### 3.5. Cholesterol Content

All SPWL fed PI-supplemented diets had significantly lower cholesterol levels, ranging from 237.1–279.7 mg/100 g lipid, than those fed the PF diet (*p* < 0.05), which had the highest cholesterol content (357.4 mg/100 g lipid) (Table 4). No significant difference in cholesterol concentration was observed among SPWL fed the F1, F2, F3, and F4 formulations (*p* > 0.05), but the cholesterol content dropped when SPWL were fed the F5 diet. These findings suggested that the type and concentration of PI influenced the total intake or in vivo biosynthesis of cholesterol in SPWL. It should be noted that feeding only GSPT resulted in the lowest cholesterol level in SPWL (*p* < 0.05). Cholesterol is a zoosterol found naturally in animals, which can be consumed through the diet or synthesized in the animal body [44]. The primary role of cholesterol is to keep cell membranes flexible and in proper alignment, as well as to act as a precursor for the production of vital substances such as vitamin D, bile acids, and steroid hormones. High dietary cholesterol intake, on the other hand, has been linked as a risk factor for hypercholesterolemia and cardiovascular/coronary artery diseases [45]. No more than 300 mg of cholesterol should be consumed daily by adults [46]. In the current study, the levels of cholesterol in SPWL fed different diet formulations could have been due to the presence of intrinsic cholesterol in the diet, particularly in PF, or to certain components in PIs or PF stimulating in vivo cholesterol biosynthesis in the corresponding SPWL. This finding was in agreement with those of Batkowska et al. [47], who noticed that supplementing hen diets with PI (linseed and SM) resulted in a decreased cholesterol level in yolk.

### 3.6. FA Profile and Health Promoting Indices of SPWL

The larval stage has a larger fat body for storing energy in the form of FAs and glucose, which is needed for a variety of processes such as metamorphosis, new adult emergence, adult flying, embriogenesis, and immune responses [27]. FAs of animal origin are typically stored as triglycerides and used for energy production via β-oxidation. Arrese and Soulages [48] suggested that feed composition influences the FA profile of insects.

Table 5 depicts the FA profile of SPWL, which was governed by the FA composition of the feed (Table 2). The FA profile of SPWL was significantly different depending on the type and concentrations of three PIs in the feed (*p* < 0.05). Palmitic acid was the most prevalent SFA in all SPWL (37.81–45.51 g/100 g lipid), while oleic acid was the most abundant UFA (34.73–41.52 g/100 g). Changes in both FAs in SPWL were strongly related to their presence in the corresponding diets (Table 2). The addition of both PI and PF supplements to the diet resulted in lower palmitic and oleic acid levels in SPWL when compared to the control diet (*p* < 0.05), resulting in the highest net SFA and MUFA contents in the corresponding SPWL. It should be noted that PI supplementation increased the levels of both essential omega-3 and omega-6 FAs in SPWL, especially linoleic acid (C18:2ω6) and α-linolenic acid (C18:3ω3). SPWL fed the F3 diet contained the greatest levels of linoleic acid (C18:2ω6), γ-linolenic acid (C18:3ω6), and arachidonic acid (C20:4), resulting in the greatest total PUFA content (*p* < 0.05). Interestingly, there was no correlation between changes in each of the three PI contents in the diets and changes in the linoleic acid content of SPWL, implying that a specific proportion of three PIs is required (Table 5). On the other hand, raising the PS content of the tested diets led to an increase in the α-linolenic acid content of SPWL, which was largely related to the amount of this FA in the tested diets (Table 2). In comparison to those fed the basal control and PF diets, the α-linolenic acid contents of SPWL fed PI-based diets were 16.80–104.98 and 4.66–47.38 times higher, respectively (Table 5). SPWL fed PI-formulated diets had increased α-linolenic acid contents, roughly 0–10.2-fold higher that those fed formulations with 0–20% PS content (Table 5), indicating that PS is a source of this essential FA. Similar trends were found in the meat of rabbits, pig, Hu lamb, and the egg yolk of *Gallus domesticus* fed diets with increased levels of PS [22,49,50,51,52].

It should be noted that PI-formulated diets had only a minor effect on the content of long-chain PUFAs in SPWL, particularly those with carbon chain lengths longer than 18 atoms, indicating a lack of FA elongation or desaturation in this insect. It was also suggested that the type and content of PUFAs in SPWL were heavily influenced by PUFA intake from the feed. The addition of PS to the feed increased the total omega-3 FA content by up to 9-fold (20% added PS in the F5 diet) compared to the PI-based diet without added PS (F1 diet), but the changes in total omega-6 FAs were different (Table 5).

SPWL fed the F3 diet provided the greatest total omega-6 FA content (8.95 g/100 g lipid), which was 4.61- and 29.88-fold higher than that in SPWL fed the PF and control diets, respectively. There were no trends in variation of omega-6 FA levels in SPWL fed PI-based diets, indicating that a specific ratio of added RB, SM, and PS in the diet was required to maximize omega-6 FA levels in the resulting SPWL. This should be explained by the fact that none of the three PIs (RB, SM, and PS) contained any dominant omega-6 FA until the presence of each of them in the diet significantly altered the net omega-6 FA content of SPWL (Table 5).

The n-6/n-3 ratio of SPWL fed PI-based diets decreased significantly with increasing PS content (*p* < 0.05). It should be noted that SPWL fed the PI-based diet without added PS (F1 diet) had a higher n-6/n-3 ratio than those fed the PF and control diets (*p* < 0.05), whereas SPWL fed the 3% PS-based diet had a comparable n-6/n-3 ratio to those fed the PF diet (*p* > 0.05). Furthermore, feeding a 7% PS-based diet reduced the n-6/n-3 ratio of SPWL to a level similar to feeding the control diet (*p* < 0.05). The addition of up to 15% PS (F4 and F5 formulations) to the tested diets resulted in lower n-6/n-3 ratios in SPWL than in those fed the PF and control diets (*p* < 0.05). Since an n-6/n-3 ratio of ≤4 is essential in reducing cardiovascular problems [22,53], feeds with more than 7% PS (F3, F4, and F5 formulations) and the control diet (C) induced greater health benefits for the lipids of SPWL. Even though feeding the control diet resulted in a lower n-6/n-3 ratio in SPWL (less than 4), there were low contents of both essential omega-3 and omega-6 fatty acids and a high content of the non-essential omega-9 fatty acid, oleic acid (Table 5). This study found that PS, as a source of linolenic acid (an omega-3 fatty acid), played an important role in lowering the n-6/n-3 ratio of SPWL. The results were in line with those of Duan et al. [52], who noticed that adding PS to chicken feed reduced the n-6/n-3 ratio in *Gallus domesticus* egg yolk. PI supplementation of the basal diet decreased the total SFA content but increased the total PUFA content of the corresponding SPWL compared to the PF and control diets (*p* < 0.05).

The total PUFA content increased with increasing PS inclusion from 3% to 20% (F2–F5 formulations) (*p* < 0.05), resulting in higher PUFA/SFA ratios of roughly 0.20–0.33 (*p* < 0.05). The PUFA/SFA index is a widely used tool for assessing how food affects cardiovascular health (CVH), with a higher ratio resulting in a lower risk of CVH [54]. PUFAs in the diet are expected to reduce LDL-C and serum cholesterol levels, whereas all SFAs are thought to contribute to high serum cholesterol levels. PS appeared to significantly alter the PUFA level of SPWL from 0.18 to 0.31 g/100 g lipid (1.72-fold) after feeding diets supplemented with 0% to 7% PS, due to an increasingly high linolenic acid content (Table 5). However, the PUFA/SFA ratio of SPWL remained constant after feeding PS-based diets with higher than 7% PS (*p* > 0.05). Our findings coincided with those of Xia et al. [50] and Arjin et al. [51], who found that as PS inclusion increased, the PUFA content increased while the SFA concentration decreased.

The IA and IT indices assess the atherogenic and thrombogenic potential of fatty acids by revealing their susceptibility to forming clots in blood vessels and determining the roles of different fatty acids. Consuming foods that induce lower IA and IT levels in human blood plasma can lower total cholesterol and LDL-C levels [54]. The IA values of all SPWL were less than 1.00, with no statistically significant difference (Table 4, *p* > 0.05). Adding PS to the basal diet, by contrast, resulted in a slight reduction in the IA, indicating that none of the three PIs played an important role in the IA of lipids extracted from SPWL. There was not a significant distinction in IT values among SPWL raised on the control, PF, and PI-based diets compared to those fed 0–3% PS-based diets (*p* > 0.05), the value of which was significantly reduced when PS inclusion levels in the diet increased to 7% (*p* < 0.05). The IT of SPWL was then statistically constant (*p* > 0.05), indicating that increasing the PS content above 7% may result in lipids in SPWL with a lower risk of blood clotting. In conclusion, supplementation with more than 7% PS (F3, F4, and F5 diets) in the GSPT diet could result in lipids from SPWL with greater health benefits.

### 3.7. Amino Acid Profile of SPWL

The quality of proteins is determined by their amino acid profile, and insect protein can be a source of essential amino acids for humans [40]. Essential amino acids are lysine, leucine, valine, isoleucine, threonine, methionine, tryptophan, phenylalanine, and histidine [41]. Interestingly, the most prevalent essential amino acid in all SPWL reared on the examined diets was lysine (*p* < 0.05), ranging from 36.4–66.2 mg/g sample (Table 6), despite being scarce in staple cereals such as rice, wheat, cassava, and maize [30]. This suggested that SPWL are a promising alternative source of lysine. Leucine was the second most abundant essential amino acid (11.3–18.4 mg/g sample), followed by valine (8.1–13.3 mg/g sample), isoleucine (6.6–10.6 mg/g sample), histidine (5.3–8.5 mg/g sample), threonine (4.2–6.5 mg/g sample), phenylalanine (3.3–5.2 mg/g sample), methionine (1.3–2.2 mg/g sample), and tryptophan (0.1–0.2 mg/g sample) (Table 6). It should be noted that feeding SPWL PI-based diets improved all nine essential amino acids when compared to the PF and control diets, suggesting that essential amino acids were derived from PIs. Among SPWL fed PI-based diets, those fed the F1 formulation containing the highest amounts of both RB and SM at 22.2% and 11.1%, respectively, and the F3 formulation containing RB, SM, and PS at moderate amounts of 17.5%, 8.8%, and 7.0%, respectively, exhibited the highest contents of nine essential amino acids. In particular, SPWL fed the F1 and F3 diets had the highest lysine levels, which were 1.75- and 1.82-fold higher than those fed the PF diet, respectively (*p* < 0.05). Furthermore, SPWL fed the F1 and F3 diets had roughly 1.60-fold higher histidine and arginine contents than those fed the PF diet (Table 6). SPWL fed the F3 diet had the highest amount of valine, at 13.3 mg/100 g sample, which was 1.64-fold greater than those fed the PF diet and slightly different from those fed the F1 diet, at 12.5 mg/100 g sample (Table 6).

Feeding SPWL the F1 and F3 diets resulted in greater amounts of five essential hydrophobic amino acids, namely valine, isoleucine, leucine, phenylalanine, and methionine, by approximately 1.64-, 11.62-, 1.57-, and 1.70-fold, respectively, compared to feeding the PF diet (Table 6). All SPWL appeared to have low tryptophan contents (0.1–0.2 mg/g sample), with SPWL fed PI-based diets having slightly higher tryptophan levels than those fed the control and PF diets (*p* < 0.05). Tryptophan is the limiting amino acid in insects, which is unavailable in *Cladomorphus phyllinum*, *Brachytrupes orientalis*, and *Tribolium castaneum* [27,40]. According to the findings, SPWL fed PI-based diets provide a good source of hydrophobic amino acids, which are considered to possess a variety of bioactive activities in terms of disease risk reduction [53]. Notably, SPWL fed the F5 diet, with the greatest amount of PS, had lesser amounts of nine essential amino acids than those fed the other PI-based diets (Table 6), indicating that PS has a minor impact on essential amino acid increment in SPWL. The inclusion of RB and SM in the tested diets, on the other hand, appeared to have a significant impact on increasing the essential amino acid content of SPWL, resulting in enhanced protein quality. This corresponded to the total EAAI and BV values being highest in SPWL raised on the F1 formulation with the highest added RB and SM amounts (Table 6). Since there was no statistically significant variation in total essential amino acid levels between SPWL fed the F1 and F3 formulations, the proper mixed ratio of RB, SM, and PS in the feed also resulted in an improved overall essential amino acid content of SPWL. SPWL fed the F1 and F3 diets had higher essential amino acid contents than *Rhynchophorus bilineatus* (40.0 mg/g), *Cladomorphus phyllinum* (83.2 mg/g), *Anoplophora chinensis* (114.4 mg/g), and *Tenebrio molitor* (124.9 mg/g) [41,53,55]. Insect essential amino acid profiles are typically thought to be comparable to that of soybean, superior to those of other vegetable proteins, and inferior to those of commercial livestock proteins [18]. In addition, EAAI is the geometrical mean of all essential amino acid contents in the studied protein compared to a highly nutritious reference protein, such as whole egg [56]. Glycine and proline were the most abundant non-essential amino acids in this insect, but other non-essential amino acids were also present in significant quantities (Table 6). SPWL fed the F3 diet had the highest non-essential amino acid level at 137.8 mg/g sample, resulting in the highest total amino acid concentration (*p* < 0.05). This corresponded strongly to the greatest protein level of this diet formulation (Table 2) and the corresponding SPWL (Table 4). According to Ghosh et al. [30], effective dietary protein consumption requires a suitable balance of essential and non-essential amino acids, as well as other nitrogen-containing molecules. In conclusion, combining RB, SM, and PS in the proper amounts (F3 formulation) with regular GSPT led to significant improvements in both protein quantity and quality of SPWL.

Table 7 compares the scores of each essential amino acid in SPWL fed different diet formulations compared to the reference protein. SPWL fed the F1 and F3 diets had the highest scores for valine, leucine, isoleucine, lysine, and total essential amino acids. The scores for other essential amino acids were greatest in SPWL fed the F1 diet (Table 7). All essential amino acids present in SPWL fed the F1 and F3 diets met FAO recommendations [43], with the exception of methionine + cysteine and tryptophan, which are the limiting essential amino acids with the lowest chemical scores in SPWL (Table 7). This also confirmed that enrichment with three PIs at appropriate concentrations (F3 diet) or high concentrations of RB and SM (F1 diet) in GSPT could improve the protein quality of this insect to comparable or even greater levels than that of the standard protein, particularly the lysine content. Our new feed formula has the potential to increase the protein quantity and quality of this insect, which will be very helpful for producing a high-quality alternative edible insect-based protein source.

### 3.8. FA Metabolism Indices and Fads2 Gene Expression

The estimated indices of fatty acid metabolism and Δ6 desaturase (*fads*2) gene expression are shown in Figure 1. Animals, including insects, do not normally produce essential long-chain fatty acids, i.e., linoleic and α-linolenic acids, from acetyl-CoA [21]. Desaturating, elongating, and terminating enzymes, in particular, have been shown to accelerate the transformation of dietary linoleic and α-linolenic acids to longer-chain PUFAs [20]. The levels of fatty acid metabolism indices and *fads*2 gene expression were examined to determine the effect of fortification with three PIs in feed on the enzymes involved in the desaturation and elongation of fatty acids in SPWL. The activity of Δ9 desaturase (18) involved in the conversion of stearic acid to oleic (18:1ω9) acid was observed at more than 90%, with the greatest activity found in SPWL fed the control and PF diets (*p* < 0.05), which was largely associated with the high concentration of n − 9 oleic acid in the corresponding SPWL (Table 4). Furthermore, SPWL fed the PF diet exhibited the greatest Δ9 desaturase (16) activity as well as the greatest total Δ9 desaturase index. Both indexes resulted in the formation of MUFA-rich palmitoleic acid (C16:1) and oleic acid in SPWL (Figure 1a). Feeding the PF and control diets to SPWL accelerated in vivo enzymes catalyzing the biosynthesis of MUFAs by adding a double bond at the ninth position (Δ9 desaturase) from the carboxyl end of fatty acids (Figure 1), resulting in increased oleic acid concentration. Furthermore, the presence of palmitoleic acid was more noticeable. This enzyme encodes a transmembrane fatty acid desaturase that uses saturated homologs to produce palmitoleic and oleic acids, similar to *Hermetia illucens* [58]. Because of the high sensitivity of the terminal thioesterase (which releases the synthesized fatty acid) to palmitic acid, the fatty acid synthase complex is known to produce large amounts of palmitic acid directly from acetyl-CoA via malonyl-CoA production. Palmitic acid is then extended to produce longer-chain fatty acids (such as stearic acid via the fatty acid elongase 6/ELOVL 6 enzyme), which are desaturated to produce UFAs or used to synthesize storage lipids [58]. The activities of the rate-limiting enzymes, Δ5 + Δ6 desaturases, determine the transformation of linoleic and α-linolenic acids to longer-chain PUFAs [20]. The activities of Δ5 + Δ6 desaturases were dramatically low in all SPWL, with those fed the control diet having the highest activities. The other SPWL did not differ in their Δ5 + Δ6 desaturase activities (*p* > 0.05), suggesting that the feed ingredient used in this study had no effect on this enzymatic activity. This insect, on the other hand, lacks Δ5 + Δ6 desaturases. Normally, the substrate specificity and catalytic efficiency of fatty acid desaturases from the same genus and even species differ. Linoleic and α-linolenic acids compete for active sites on Δ5 + Δ6 desaturase enzymes when both are present, although their affinities are different [59]. SPWL fed the control diet had the greatest thioesterase activity (*p* < 0.05), but no difference in thioesterase activity was found among SPWL fed PI-supplemented and PF diets (*p* > 0.05). Since thioesterase activity was low in all SPWL, fatty acid biosynthesis to longer 18-carbon chains was limited. This was related to the fatty acid profile of SPWL (Table 5).

Figure 1b depicts the effects of PI-added diets on *fads*2 gene expression in SPWL. The *fads*2 mRNA expression followed the same pattern as the Δ5 + Δ6 desaturase index. The supplementation of any PI to a regular diet lowered the expression of the *fads*2 gene in SPWL when compared to the control diet (*p* < 0.05). There was no notable variation in *fads*2 gene expression among SPWL fed the F1, F2, F3, and F4 diets (*p* > 0.05), but the level was slightly increased in those fed the F5 diet (highest PS content of 20%) (*p* < 0.05). It should be noted that extremely low amounts of long-chain PUFAs with more than 18 carbons, particularly EPA and DHA, were found in all SPWL fed various diets (Table 4), which was highly related to the low levels of all FA metabolism indices and *fads*2 gene expression. According to Mattioli et al. [60], despite high amounts of ALA added in feeds, *Tenebrio molitor* larvae were virtually unable to generate long-chain fatty acids de novo. The PUFAs in the feed were simply bioaccumulated by the larvae, which converted roughly two-thirds of PUFAs into SFAs such as lauric or myristic acid. In conclusion, diet composition had no effect on the FA-related enzymes of SPWL, and the presence of FAs in the larvae was primarily attributed to the FA composition of the diet. This also confirmed the critical role of FAs found in PIs for improving essential FA concentrations in SPWL.

## 4. Conclusions

The diet composition is critical in determining the growth performance and nutritional value of SPWL. The balance of RB, SM, and PS with GSPT in the diet resulted in improved protein and lipid quantities and qualities in SPWL, as well as increased macro- and micromineral levels, which stimulated growth and induced a higher survival rate than the commercial PF and regular GSPT diets. In comparison to the commercial PF and typical GSPT diets, PIs increased the yield expressed as live weight up to two-fold. Furthermore, optimizing the feed composition can lead to increased production efficiency and sustainability of farm-raised SPWL production. Overall, incorporating PIs into a regular diet is a promising strategy for improving the nutritional quality and sustainability of farm-raised SPWL as a potential alternative protein and lipid source.

## Figures and Tables

**Figure 1 foods-12-03474-f001:**
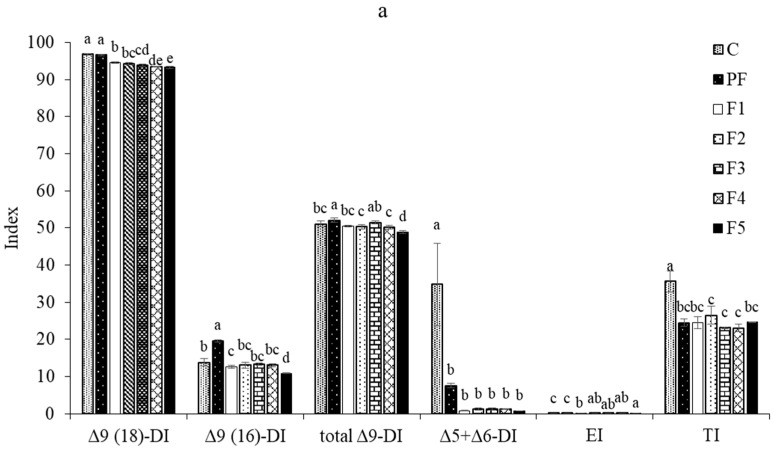
Fatty acid metabolism indices (**a**) and *fads*2 gene expression (**b**) of sago palm weevil larvae raised on different plant-based ingredient-supplemented diet formulations. Bars indicate standard deviation from triplicate determinations. Significant differences (*p* < 0.05) are indicated by different letters in the same attribute. Experimental diet nomenclature: See caption in Table 1. DI, desaturase index; EI, elongase index; TI, thioesterase index.

**Table 1 foods-12-03474-t001:** Composition of feed formulations.

Ingredient (%)	C	PF	F1	F2	F3	F4	F5
Ground sago palm trunk (GSPT)	100	66.7	66.7	66.7	66.7	66.7	66.7
Commercial pig feed (PF)		33.3					
Rice bran (RB)			22.2	20.2	17.5	12.2	8.9
Soyben meal (SM)			11.1	10.1	8.8	6.1	4.4
Perilla seed (PS)			0	3.0	7.0	15.0	20.0

**Table 2 foods-12-03474-t002:** Proximate composition and fatty acid profile of experimental diets.

Composition	C	PF	F1	F2	F3	F4	F5
Proximate composition							
Moisture (g/100 g)	68.8 ± 0.7 ^a^	53.1 ± 1.1 ^f^	60.2 ± 0.2 ^b^	58.7 ± 0.3 ^c^	57.2 ± 0.2 ^d^	53.8 ± 0.3 ^e^	54.5 ± 0.9 ^e^
Protein (g/100 g, dw)	1.0 ± 0.0 ^d^	11.3 ± 0.1 ^c^	13.4 ± 0.0 ^a^	12.5 ± 0.5 ^b^	12.5 ± 0.6 ^b^	11.5 ± 0.1 ^c^	12.5 ± 0.1 ^b^
Lipid (g/100 g, dw)	0.8 ± 0.1 ^d^	2.0 ± 0.0 ^d^	6.2 ± 0.5 ^c^	7.3 ± 0.6 ^c^	16.1 ± 1.4 ^b^	17.3 ± 2.0 ^b^	21.0 ± 1.2 ^a^
Ash (g/100 g, dw)	4.1 ± 0.0 ^g^	8.1 ± 0.1 ^a^	5.5 ± 0.0 ^b^	5.2 ± 0.0 ^c^	4.8 ± 0.1 ^d^	4.5 ± 0.0 ^e^	4.3 ± 0.2 ^f^
Carbohydrate (g/100 g, dw)	94.1 ± 0.2 ^a^	78.6 ± 0.0 ^b^	74.9 ± 0.5 ^c^	75.0 ± 0.9 ^c^	66.7 ± 1.4 ^d^	66.6 ± 1.9 ^d^	62.2 ± 1.3 ^e^
Fatty acid (%total lipid)							
C12:0	0.87 ± 0.13 ^b^	5.12 ± 0.18 ^a^	0.11 ± 0.01 ^c^	0.10 ± 0.02 ^c^	0.09 ± 0.01 ^c^	0.08 ± 0.01 ^c^	0.09 ± 0.02 ^c^
C14:0	1.99 ± 0.16 ^a^	1.86 ± 007 ^a^	0.54 ± 0.01 ^b^	0.45 ± 0.06 ^bc^	0.42 ± 0.05 ^bc^	0.33 ± 0.07 ^c^	0.33 ± 0.03 ^c^
C14:1	0.27 ± 0.17 ^a^	0.04 ± 0.03 ^b^	0.01 ± 0.01 ^b^	0.02 ± 0.01 ^b^	0.03 ± 0.01 ^b^	0.07 ± 0.08 ^b^	0.04 ± 0.04 ^b^
C15:0	0.45 ± 0.02 ^a^	0.11 ± 0.02 ^b^	0.07 ± 0.01 ^c^	0.06 ± 0.01 ^c^	0.06 ± 0.01 ^c^	0.05 ± 0.01 ^c^	0.05 ± 0.01 ^c^
C15:1 (cis-10)	0.37 ± 0.25 ^ns^	0.19 ± 0.04 ^ns^	0.07 ± 0.02 ^ns^	0.13 ± 0.10 ^ns^	0.10 ± 0.04 ^ns^	0.07 ± 0.04 ^ns^	0.12 ± 0.06 ^ns^
C16:0	34.51 ± 0.78 ^a^	21.96 ± 0.12 ^b^	22.47 ± 0.30 ^b^	18.96 ± 0.91 ^c^	16.51 ± 0.41 ^d^	13.54 ± 0.87 ^e^	11.79 ± 0.21 ^f^
C16:1 (cis-9)	0.83 ± 0.27 ^a^	0.42 ± 0.14 ^b^	0.18 ± 0.03 ^b^	0.16 ± 0.04 ^b^	0.25 ± 0.06 ^b^	0.18 ± 0.08 ^b^	0.19 ± 0.12 ^b^
C17:0	0.51 ± 0.03 ^a^	0.12 ± 0.01 ^b^	0.11 ± 0.02 ^bc^	0.09 ± 0.01 ^c^	0.10 ± 0.01 ^bc^	0.09 ± 0.01 ^c^	0.08 ± 0.01 ^c^
C17:1 (cis-10)	0.23 ± 0.04 ^a^	0.02 ± 0.01 ^b^	0.07 ± 0.03 ^b^	0.02 ± 0.02 ^b^	0.04 ± 0.03 ^b^	0.04 ± 0.04 ^b^	0.03 ± 0.02 ^b^
C18:0	9.76 ± 0.57 ^a^	3.31 ± 0.08 ^c^	3.96 ± 0.11 ^b^	4.05 ± 0.07 ^b^	3.94 ± 0.10 ^b^	3.93 ± 0.10 ^b^	3.62 ± 0.07 ^bc^
C18:1 (cis-9)	13.39 ± 0.61 ^f^	24.74 ± 0.29 ^c^	32.19 ± 1.15 ^a^	25.93 ± 0.57 ^b^	20.17 ± 0.63 ^d^	15.24 ± 0.51 ^e^	12.28 ± 0.18 ^g^
C18:2 (all-cis-9,12)	5.43 ± 0.48 ^g^	36.73 ± 0.36 ^a^	35.05 ± 1.79 ^b^	30.66 ± 0.93 ^c^	25.98 ± 0.20 ^d^	22.41 ± 0.71 ^e^	20.16 ± 0.22 ^f^
C18:3 (all-cis-cis -6,9,12)	0.21 ± 0.13 ^a^	0.04 ± 0.04 ^b^	0.04 ± 0.04 ^b^	0.01 ± 0.01 ^b^	0.02 ± 0.03 ^b^	0.05 ± 0.01 ^b^	0.02 ± 0.01 ^b^
C18:3 (all-cis-cis -9,12, 15)	1.83 ± 0.17 ^ef^	1.38 ± 0.01 ^f^	2.34 ± 0.14 ^e^	16.85 ± 0.70 ^d^	29.64 ± 0.41 ^c^	41.32 ± 0.72 ^b^	48.84 ± 0.44 ^a^
C20:0	0.37 ± 0.09 ^a^	0.26 ± 0.04 ^b^	0.39 ± 0.01 ^a^	0.34 ± 0.02 ^a^	0.27 ± 0.01 ^b^	0.23 ± 0.02 ^bc^	0.19 ± 0.01 ^c^
C20:3 (all-cis-8,11,14)	0.31 ± 0.25 ^a^	0.04 ± 0.03 ^b^	0.06 ± 0.02 ^b^	0.04 ± 0.04 ^b^	0.02 ± 0.00 ^b^	0.04 ± 0.01 ^b^	0.08 ± 0.00 ^b^
C20:4n-6	0.17 ± 0.13 ^ns^	0.02 ± 0.01 ^ns^	0.02 ± 0.01 ^ns^	0.06 ± 0.02 ^ns^	0.05 ± 0.03 ^ns^	0.004 ± 0.06 ^ns^	0.02 ± 0.01 ^ns^
C20:3 (all-cis-11,14,17)	0.58 ± 0.09 ^a^	0.08 ± 0.06 ^b^	0.04 ± 0.04 ^b^	0.01 ± 0.01 ^b^	0.06 ± 0.03 ^b^	0.05 ± 0.01 ^b^	0.04 ± 0.03 ^b^
C20:5 (all -cis-5,8,11,14,17)	0.10 ± 0.01 ^a^	0.05 ± 0.04 ^b^	0.06 ± 0.03 ^b^	0.03 ± 0.01 ^b^	0.03 ± 0.03 ^b^	0.02 ± 0.01 ^b^	0.03 ± 0.00 ^b^
C22:2 (all-cis-13,16)	0.48 ± 0.01 ^a^	0.06 ± 0.04 ^b^	0.03 ± 0.03 ^b^	0.04 ± 0.04 ^b^	0.06 ± 0.02 ^b^	0.08 ± 0.10 ^b^	0.10 ± 0.03 ^b^
C22:6 (all-cis-4,7,10,13,16,19)	0.11 ± 0.07 ^ns^	0.07 ± 0.04 ^ns^	0.04 ± 0.03 ^ns^	0.04 ± 0.04 ^ns^	0.03 ± 0.02 ^ns^	0.03 ± 0.03 ^ns^	0.04 ± 0.04 ^ns^
other	27.23 ± 1.78 ^a^	3.37 ± 0.04 ^b^	2.15 ± 0.61 ^bc^	1.94 ± 0.19 ^bc^	2.13 ± 0.36 ^bc^	2.07 ± 0.57 ^bc^	1.82 ± 0.29 ^c^
∑SFA	48.45 ± 1.34 ^a^	32.74 ± 0.29 ^b^	27.65 ± 0.20 ^c^	24.05 ± 1.09 ^d^	21.38 ± 0.53 ^e^	18.26 ± 1.08 ^f^	16.15 ± 0.31 ^g^
∑MUFA	15.08 ± 0.69 ^d^	25.40 ± 0.22 ^b^	32.52 ± 1.17 ^a^	26.27 ± 0.71 ^b^	20.59 ± 0.66 ^c^	15.59 ± 0.35 ^d^	12.66 ± 0.22 ^e^
∑PUFA	9.04 ± 0.36 ^e^	38.76 ± 0.65 ^d^	42.30 ± 7.98 ^d^	52.42 ± 8.09 ^c^	59.36 ± 6.32 ^bc^	66.68 ± 4.13 ^ab^	69.42 ± 0.33 ^a^
∑n-3	0.28 ± 0.02 ^e^	0.80 ± 0.04 ^e^	1.54 ± 0.16 ^e^	10.28 ± 1.71 ^d^	17.54 ± 0.63 ^c^	26.44 ± 4.80 ^b^	32.00 ± 0.57 ^a^
∑n-6	6.13 ± 0.22 ^g^	36.84 ± 0.29 ^a^	35.17 ± 1.75 ^b^	30.76 ± 0.87 ^c^	26.08 ± 0.22 ^d^	22.54 ± 0.64 ^e^	20.28 ± 0.24 ^f^
n-6/n-3	21.63 ± 1.33 ^b^	46.26 ± 2.59 ^a^	22.89 ± 1.37 ^b^	3.04 ± 0.47 ^c^	1.49 ± 0.04 ^cd^	0.87 ± 0.15 ^cd^	0.63 ± 0.02 ^d^

Values are presented as the mean ± standard deviation based on triplicate determinations. Different letters in the same row indicate significant differences (*p* < 0.05). ns, non-significant differences. Experimental diet nomenclature: See Table 1.

**Table 3 foods-12-03474-t003:** Growth performance of sago palm weevil larvae fed different diets compared to the reference protein.

Biometric Parameter	C	PF	F1	F2	F3	F4	F5
Survival rate (%)	89.2 ± 1.2 ^ab^	90.0 ± 0.5 ^ab^	89.3 ± 0.0 ^ab^	90.6 ± 0.6 ^a^	90.5 ± 0.4 ^a^	88.5 ± 1.0 ^bc^	87.2 ± 0.2 ^c^
Live weight (g)	2.6 ± 1.3 ^d^	5.2 ± 0.5 ^b^	5.1 ± 0.6 ^b^	5.9 ± 0.7 ^a^	4.3 ± 0.3 ^c^	4.6 ± 0.4 ^c^	5.9 ± 0.5 ^a^
Dry weight (g)	0.9 ± 0.0 ^c^	1.8 ± 0.1 ^b^	1.6 ± 0.2 ^b^	2.3 ± 0.3 ^a^	1.2 ± 0.1 ^c^	1.6 ± 0.2 ^b^	2.4 ± 0.2 ^a^
Dry matter content (%)	27.5 ± 1.1 ^d^	31.2 ± 2.4 ^cd^	33.6 ± 2.1 ^bc^	34.8 ± 2.8 ^bc^	29.0 ± 2.0 ^d^	36.8 ± 2.2 ^b^	42.5 ± 2.0 ^a^
Condition factor	8.1 ± 3.0 ^c^	9.5 ± 2.5 ^bc^	11.6 ± 2.3 ^a^	11.9 ± 2.1 ^a^	12.0 ± 2.1 ^a^	10.5 ± 2.9 ^ab^	9.5 ± 1.4 ^bc^
Growth rate (g/day)	0.11 ± 0.07 ^b^	0.20 ± 0.05 ^a^	0.05 ± 0.05 ^cd^	0.07 ± 0.05 ^cd^	0.04 ± 0.03 ^d^	0.05 ± 0.03 ^d^	0.08 ± 0.03 ^c^
Appearance	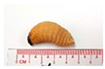	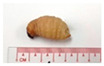	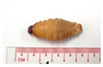	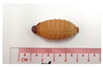	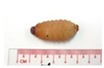	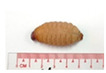	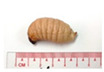

Values are presented as the mean ± standard deviation of determinations made in triplicate. Significant differences (*p* < 0.05) are indicated by different letters in the same row. Experimental diet nomenclature: See Table 1.

**Table 4 foods-12-03474-t004:** Proximate composition, cholesterol content, and mineral composition of sago palm weevil larvae fed different diets.

Sample	C	PF	F1	F2	F3	F4	F5
Moisture (g/100 g)	72.6 ± 1.1 ^a^	71.1 ± 0.6 ^a^	66.4 ± 2.1 ^b^	65.2 ± 2.8 ^b^	71.0 ± 2.0 ^a^	63.2 ± 2.2 ^b^	57.5 ± 2.0 ^c^
Protein (g/100 g, dw)	22.2 ± 0.3 ^cd^	18.7 ± 0.0 ^e^	24.4 ± 0.4 ^b^	23.8 ± 0.7 ^b^	26.8 ± 1.5 ^a^	23.2 ± 0.6 ^bc^	21.4 ± 0.8 ^d^
Lipid (g/100 g, dw)	33.0 ± 0.4 ^e^	57.3 ± 1.4 ^ab^	50.3 ± 1.6 ^c^	58.1 ± 1.8 ^a^	54.1 ± 2.4 ^b^	54.5 ± 3.3 ^ab^	44.0 ± 2.5 ^d^
Ash (g/100 g, dw)	4.2 ± 0.0 ^a^	3.0 ± 0.2 ^bc^	2.7 ± 0.2 ^cd^	3.0 ± 0.3 ^bc^	3.2 ± 0.2 ^b^	2.6 ± 0.2 ^de^	2.2 ± 0.1 ^e^
Carbohydrate (g/100 g, dw)	40.6 ± 0.1 ^a^	21.0 ± 1.1 ^c^	22.6 ± 1.7 ^c^	15.1 ± 1.2 ^d^	15.9 ± 2.3 ^d^	19.7 ± 3.1 ^c^	32.4 ± 3.0 ^b^
Cholesterol (mg/100 g lipid)	246.9 ± 0.7 ^d^	357.4 ± 6.6 ^a^	237.1 ± 4.7 ^b^	279.7 ± 1.7 ^b^	277.2 ± 2.1 ^b^	277.2 ± 1.4 ^b^	264.7 ± 4.4 ^c^
Mineral (mg/kg)							
Potassium (K)	9417 ± 111 ^a^	6461 ± 9 ^d^	7116 ± 135 ^c^	6384 ± 135 ^d^	8220 ± 188 ^b^	6094 ± 243 ^e^	5652 ± 31 ^f^
Phosphorus (P)	3557 ± 112 ^cd^	2908 ± 46 ^e^	3721 ± 68 ^b^	3232 ± 30 ^d^	4003 ± 90 ^a^	3182 ± 84 ^d^	2869 ± 21 ^e^
Magnesium (Mg)	1845 ± 28 ^b^	1310 ± 11 ^f^	1763 ± 25 ^c^	1460 ± 28 ^e^	2082 ± 37 ^a^	1543 ± 46 ^d^	1240 ± 12 ^g^
Sodium (Na)	1331 ± 9 ^a^	980 ± 36 ^c^	1082 ± 4 ^b^	889 ± 23 ^d^	1374 ± 50 ^a^	1003 ± 27 ^c^	827 ± 9 ^e^
Calcium (Ca)	659 ± 13 ^b^	431 ± 3 ^f^	592 ± 13 ^c^	491 ± 14 ^e^	731 ± 20 ^a^	537 ± 8 ^d^	452 ± 3 ^f^
Zinc (Zn)	98.6 ± 1.7 ^c^	101.4 ± 1.1 ^b^	95.2 ± 1.8 ^d^	80.6 ± 0.3 ^f^	131.6 ± 1.4 ^a^	87.6 ± 1.1 ^e^	78.1 ± 0.9 ^g^
Manganese (Mn)	49.4 ± 1.2 ^a^	13.4 ± 0.3 ^c^	13.3 ± 0.1 ^c^	11.5 ± 0.3 ^d^	21.3 ± 0.2 ^b^	13.9 ± 0.3 ^c^	9.6 ± 0.1 ^e^
Iron (Fe)	14.0 ± 0.1 ^d^	15.2 ± 0.1 ^c^	17.5 ± 0.3 ^b^	15.1 ± 0.2 ^c^	18.8 ± 0.6 ^a^	14.5 ± 0.5 ^cd^	13.2 ± 0.5 ^e^
Copper (Cu)	10.5 ± 0.4 ^de^	25.9 ± 0.4 ^a^	11.2 ± 0.8 ^d^	10.0 ± 0.1 ^e^	14.5 ± 0.4 ^b^	12.9 ± 0.4 ^c^	11.1 ± 0.5 ^d^

Values are presented as the mean ± standard deviation of determinations made in triplicate. Significant differences (*p* < 0.05) are indicated by different letters in the same row. Experimental diet nomenclature: See Table 1.

**Table 5 foods-12-03474-t005:** Fatty acid composition (g/100 g total lipid) of sago palm weevil larvae fed different diets.

Fatty Acid (g/100 g Total Lipid)	C	PF	F1	F2	F3	F4	F5
C12:0	0.12 ± 0.04 ^ns^	0.16 ± 0.02 ^ns^	0.13 ± 0.02 ^ns^	0.12 ± 0.02 ^ns^	0.14 ± 0.02 ^ns^	0.15 ± 0.01 ^ns^	0.09 ± 0.07 ^ns^
C14:0	1.28 ± 0.07 ^c^	1.77 ± 0.08 ^a^	1.70 ± 0.11 ^ab^	1.56 ± 0.14 ^b^	1.64 ± 0.01 ^ab^	1.69 ± 0.06 ^ab^	1.58 ± 0.00 ^b^
C14:1	0.04 ± 0.01 ^ns^	0.11 ± 0.08 ^ns^	0.08 ± 0.08 ^ns^	0.10 ± 0.04 ^ns^	0.03 ± 0.02 ^ns^	0.05 ± 0.04 ^ns^	0.04 ± 0.01 ^ns^
C15:0	0.05 ± 0.01 ^ns^	0.07 ± 0.01 ^ns^	0.07 ± 0.01 ^ns^	0.07 ± 0.01 ^ns^	0.10 ± 0.01 ^ns^	0.07 ± 0.04 ^ns^	0.09 ± 0.00 ^ns^
C15:1 (cis-10)	0.33 ± 0.08 ^b^	0.16 ± 0.20 ^b^	0.16 ± 0.01 ^b^	0.13 ± 0.09 ^b^	0.88 ± 0.33 ^a^	1.02 ± 0.28 ^a^	0.94 ± 0.15 ^a^
C16:0	45.51 ± 1.14 ^a^	43.35 ± 0.28 ^b^	41.41 ± 0.05 ^c^	41.24 ± 0.96 ^c^	37.81 ± 0.30 ^d^	38.87 ± 0.25 ^d^	38.89 ± 0.15 ^d^
C16:1 (cis-9)	7.27 ± 0.48 ^b^	10.58 ± 0.25 ^a^	6.02 ± 0.21 ^c^	6.18 ± 0.38 ^c^	5.81 ± 0.16 ^c^	5.82 ± 0.20 ^c^	4.73 ± 0.08 ^d^
C17:0	0.03 ± 0.03 ^ns^	0.04 ± 0.04 ^ns^	0.07 ± 0.03 ^ns^	0.08 ± 0.01 ^ns^	0.09 ± 0.00 ^ns^	0.08 ± 0.01 ^ns^	0.07 ± 0.03 ^ns^
C17:1 (cis-10)	0.06 ± 0.01 ^ns^	0.07 ± 0.05 ^ns^	0.03 ± 0.00 ^ns^	0.04 ± 0.01 ^ns^	0.03 ± 0.03 ^ns^	0.02 ± 0.02 ^ns^	0.03 ± 0.02 ^ns^
C18:0	1.34 ± 0.08 ^d^	1.33 ± 0.03 ^d^	2.23 ± 0.09 ^c^	2.30 ± 0.15 ^bc^	2.40 ± 0.08 ^ab^	2.48 ± 0.04 ^a^	2.49 ± 0.07 ^a^
C18:1 (cis-9)	41.52 ± 0.24 ^a^	37.90 ± 1.21 ^bc^	38.56 ± 0.56 ^b^	37.90 ± 0.83 ^bc^	36.81 ± 0.63 ^cd^	35.77 ± 0.78 ^de^	34.73 ± 0.51 ^e^
C18:2 (all-cis-9,12)	0.14 ± 0.08 ^e^	1.72 ± 0.12 ^d^	7.03 ± 0.18 ^b^	6.83 ± 0.24 ^b^	8.47 ± 0.41 ^a^	5.86 ± 0.24 ^c^	5.49 ± 0.20 ^c^
C18:3 (all-cis-cis-6,9,12)	0.01 ± 0.00 ^c^	0.03 ± 0.00 ^bc^	0.05 ± 0.03 ^bc^	0.04 ± 0.05 ^bc^	0.27 ± 0.03 ^a^	0.09 ± 0.06 ^b^	0.06 ± 0.04 ^bc^
C18:3 (all-cis-cis-9,12, 15)	0.05 ± 0.01 ^f^	0.18 ± 0.01 ^f^	0.84 ± 0.12 ^e^	1.52 ± 0.15 ^d^	3.35 ± 0.21 ^c^	5.74 ± 0.23 ^b^	8.53 ± 0.27 ^a^
C20:0	0.11 ± 0.01 ^a^	0.07 ± 0.02 ^c^	0.09 ± 0.02 ^abc^	0.11 ± 0.02 ^ab^	0.08 ± 0.01 ^c^	0.08 ± 0.00 ^bc^	0.09 ± 0.01 ^abc^
C20:3 (all-cis-8,11,14)	0.14 ± 0.11 ^ns^	0.18 ± 0.10 ^ns^	0.03 ± 0.03 ^ns^	0.05 ± 0.01 ^ns^	0.14 ± 0.10 ^ns^	0.05 ± 0.03 ^ns^	0.03 ± 0.02 ^ns^
C20:4n-6	0.01 ± 0.01 ^b^	0.02 ± 0.00 ^b^	0.02 ± 0.00 ^b^	0.05 ± 0.01 ^ab^	0.07 ± 0.04 ^a^	0.02 ± 0.01 ^b^	0.01 ± 0.00 ^b^
C20:3 (all-cis-11,14,17)	0.04 ± 0.03 ^ns^	0.12 ± 0.08 ^ns^	0.08 ± 0.06 ^ns^	0.03 ± 0.01 ^ns^	0.06 ± 0.04 ^ns^	0.05 ± 0.04 ^ns^	0.04 ± 0.02 ^ns^
C20:5 (all-cis-5,8,11,14,17)	0.05 ± 0.01 ^b^	0.08 ± 0.01 ^a^	0.02 ± 0.00 ^cd^	0.02 ± 0.00 ^cd^	0.03 ± 0.01 ^c^	0.05 ± 0.00 ^b^	0.02 ± 0.00 ^d^
C22:2 (all-cis-13,16)	0.03 ± 0.04 ^ns^	0.05 ± 0.02 ^ns^	0.05 ± 0.01 ^ns^	0.05 ± 0.02 ^ns^	0.08 ± 0.06 ^ns^	0.06 ± 0.01 ^ns^	0.07 ± 0.00 ^ns^
C22:6 (all-cis-4,7,10,13,16,19)	0.03 ± 0.01 ^de^	0.06 ± 0.01 ^bc^	0.02 ± 0.01 ^f^	0.03 ± 0.01 ^ef^	0.05 ± 0.01 ^cd^	0.08 ± 0.00 ^a^	0.07 ± 0.00 ^ab^
other	1.82 ± 0.21 ^ns^	1.94 ± 0.41 ^ns^	1.31 ± 0.20 ^ns^	1.54 ± 0.16 ^ns^	1.68 ± 0.15 ^ns^	1.91 ± 0.68 ^ns^	1.90 ± 0.77 ^ns^
∑SFA	48.45 ± 0.95 ^a^	46.80 ± 0.37 ^b^	45.70 ± 0.16 ^c^	45.49 ± 0.78 ^c^	42.26 ± 0.35 ^e^	43.42 ± 0.10 ^d^	43.30 ± 0.18 ^d^
∑MUFA	49.22 ± 0.52 ^a^	48.83 ± 0.83 ^a^	44.85 ± 0.38 ^b^	44.36 ± 0.42 ^bc^	43.56 ± 0.77 ^cd^	42.69 ± 0.45 ^d^	40.46 ± 0.35 ^e^
∑PUFA	0.55 ± 0.33 ^d^	2.63 ± 0.20 ^c^	8.38 ± 0.23 ^b^	9.29 ± 0.87 ^b^	13.18 ± 0.75 ^a^	13.03 ± 1.60 ^a^	14.39 ± 0.28 ^a^
∑n-3	0.17 ± 0.05 ^f^	0.44 ± 0.07 ^f^	0.96 ± 0.18 ^e^	1.60 ± 0.17 ^d^	3.49 ± 0.25 ^c^	5.91 ± 0.19 ^b^	8.66 ± 0.24 ^a^
∑n-6	0.30 ± 0.17 ^f^	1.94 ± 0.04 ^e^	7.14 ± 0.12 ^b^	6.97 ± 0.27 ^b^	8.95 ± 0.45 ^a^	6.02 ± 0.17 ^c^	5.59 ± 0.19 ^d^
n-6/n-3	1.68 ± 0.48 ^cd^	4.51 ± 0.79 ^b^	7.63 ± 1.31 ^a^	4.39 ± 0.29 ^b^	2.57 ± 0.08 ^c^	1.02 ± 0.00 ^d^	0.65 ± 0.00 ^d^
PUFA/SFA	0.01 ± 0.01 ^e^	0.06 ± 0.01 ^d^	0.18 ± 0.01 ^c^	0.20 ± 0.02 ^c^	0.31 ± 0.02 ^ab^	0.30 ± 0.04 ^b^	0.33 ± 0.01 ^a^
IA	0.53 ± 0.12 ^ns^	0.45 ± 0.04 ^ns^	0.50 ± 0.07 ^ns^	0.51 ± 0.11 ^ns^	0.41 ± 0.04 ^ns^	0.38 ± 0.02 ^ns^	0.44 ± 0.01 ^ns^
IT	0.94 ± 0.22 ^a^	0.79 ± 0.07 ^ab^	0.86 ± 0.14 ^a^	0.84 ± 0.19 ^a^	0.59 ± 0.07 ^bc^	0.45 ± 0.02 ^c^	0.45 ± 0.01 ^c^

Values are presented as the mean ± standard deviation of determinations made in triplicate. Significant differences (*p* < 0.05) are indicated by different letters in the same row. ns, non-significant differences. Experimental diet nomenclature: See Table 1. IA, index of atherogenicity. IT, index of thrombogenicity.

**Table 6 foods-12-03474-t006:** Amino acid profile of sago palm weevil larvae fed different diets.

Amino Acid (mg/g Sample)	C	PF	F1	F2	F3	F4	F5
Essential amino acid (EAA)							
Valine	9.3 ± 0.1 ^d^	8.1 ± 1.0 ^e^	12.5 ± 0.3 ^b^	10.4 ± 0.2 ^c^	13.3 ± 0.2 ^a^	10.7 ± 0.2 ^c^	10.0 ± 0.3 ^cd^
Leucine	13.2 ± 0.1 ^d^	11.3 ± 1.4 ^e^	17.9 ± 0.5 ^a^	14.5 ± 0.3 ^bc^	18.4 ± 0.3 ^a^	15.3 ± 0.2 ^b^	13.8 ± 0.5 ^cd^
Isoleucine	7.5 ± 0.2 ^c^	6.6 ± 0.8 ^d^	10.2 ± 0.2 ^a^	8.4 ± 0.1 ^b^	10.6 ± 0.1 ^a^	8.7 ± 0.2 ^b^	8.0 ± 0.2 ^bc^
Threonine	4.8 ± 0.1 ^d^	4.2 ± 0.4 ^e^	6.3 ± 0.1 ^a^	5.2 ± 0.0 ^bc^	6.5 ± 0.1 ^a^	5.4 ± 0.1 ^b^	5.0 ± 0.1 ^cd^
Lysine	43.9 ± 1.1 ^c^	36.4 ± 4.9 ^d^	63.7 ± 2.0 ^a^	47.6 ± 1.3 ^c^	66.2 ± 1.3 ^a^	52.3 ± 1.5 ^b^	47.6 ± 1.7 ^c^
Histidine	6.0 ± 0.1 ^d^	5.3 ± 0.6 ^e^	8.3 ± 0.2 ^a^	6.9 ± 0.1 ^bc^	8.5 ± 0.1 ^a^	7.1 ± 0.1 ^b^	6.5 ± 0.1 ^c^
Tryptophan	0.1 ± 0.0 ^c^	0.1 ± 0.0 ^c^	0..2 ± 0.0 ^a^	0.2 ± 0.0 ^b^	0.2 ± 0.0 ^a^	0.2 ± 0.0 ^b^	0.1 ± 0.0 ^c^
Phenylalanine	3.8 ± 0.0 ^d^	3.3 ± 0.4 ^e^	5.2 ± 0.1 ^a^	4.2 ± 0.1 ^bc^	5.2 ± 0.1 ^a^	4.3 ± 0.1 ^b^	4.0 ± 0.1 ^cd^
Methionine	1.5 ± 0.0 ^c^	1.3 ± 0.2 ^d^	2.1 ± 0.1 ^a^	1.7 ± 0.0 ^c^	2.2 ± 0.1 ^a^	1.8 ± 0.0 ^b^	1.6 ± 0.0 ^c^
Total EAA	90.0 ± 1.4 ^d^	76.5 ± 9.6 ^e^	126.5 ± 3.0 ^a^	99.0 ± 1.2 ^bc^	131.1 ± 2.1 ^a^	105.7 ± 2.0 ^b^	96.7 ± 3.0 ^cd^
EAAI (%)	44.8 ± 0.5 ^d^	46.6 ± 4.8 ^cd^	56.2 ± 1.3 ^a^	46.3 ± 0.5 ^cd^	52.4 ± 0.6 ^b^	49.6 ± 0.6 ^bc^	48.6 ± 1.3 ^c^
BV (%)	37.1 ± 0.5 ^d^	39.1 ± 5.3 ^cd^	49.5 ± 1.4 ^a^	38.7 ± 0.5 ^cd^	45.5 ± 0.7 ^b^	42.4 ± 0.7 ^bc^	41.3 ± 1.4 ^c^
Non-essential amino acid							
Cysteine	0.0 ± 0.0 ^ns^	0.0 ± 0.0 ^ns^	0.0 ± 0.0 ^ns^	0.0 ± 0.0 ^ns^	0.0 ± 0.0 ^ns^	0.0 ± 0.0 ^ns^	0.0 ± 0.0 ^ns^
Tyrosine	3.8 ± 0.1 ^d^	3.4 ± 0.2 ^e^	5.0 ± 0.1 ^a^	4.1 ± 0.1 ^bc^	5.2 ± 0.2 ^a^	4.3 ± 0.1 ^b^	4.0 ± 0.1 ^cd^
Alanine	9.5 ± 0.0 ^bc^	7.8 ± 0.8 ^d^	12.4 ± 0.3 ^a^	10.1 ± 0.2 ^b^	12.6 ± 0.2 ^a^	10.0 ± 0.2 ^b^	9.3 ± 0.2 ^c^
Glycine	17.4 ± 0.3 ^d^	15.1 ± 1.7 ^e^	25.0 ± 0.6 ^a^	20.9 ± 0.5 ^b^	26.2 ± 0.1 ^a^	21.1 ± 0.4 ^b^	19.5 ± 0.5 ^c^
Serine	3.2 ± 0.1 ^c^	2.9 ± 0.3 ^d^	4.1 ± 0.1 ^a^	3.4 ± 0.1 ^bc^	4.3 ± 0.2 ^a^	3.7 ± 0.1 ^b^	3.4 ± 0.0 ^c^
Aspartic acid	5.0 ± 0.1 ^d^	4.5 ± 0.3 ^e^	6.6 ± 0.2 ^a^	5.5 ± 0.1 ^bc^	6.8 ± 0.3 ^a^	5.7 ± 0.1 ^b^	5.2 ± 0.0 ^cd^
Hydroxyproline	0.4 ± 0.0 ^b^	0.4 ± 0.0 ^c^	0.5 ± 0.0 ^a^	0.4 ± 0.0 ^b^	0.5 ± 0.0 ^a^	0.4 ± 0.0 ^bc^	0.4 ± 0.0 ^bc^
Proline	13.5 ± 0.3 ^d^	12.8 ± 1.5 ^d^	22.1 ± 0.4 ^b^	18.2 ± 0.3 ^c^	23.4 ± 0.3 ^a^	17.7 ± 0.3 ^c^	18.7 ± 0.6 ^c^
Glutamic acid	9.7 ± 0.2 ^d^	8.6 ± 0.6 ^e^	12.6 ± 0.2 ^b^	10.2 ± 0.2 ^d^	13.2 ± 0.4 ^a^	11.0 ± 0.2 ^c^	10.1 ± 0.2 ^d^
Arginine	10.3 ± 0.0 ^cd^	8.1 ± 0.8 ^f^	13.4 ± 0.2 ^a^	9.9 ± 0.0 ^de^	12.6 ± 0.4 ^b^	10.9 ± 0.4 ^c^	9.5 ± 0.2 ^e^
Ornithine	23.4 ± 0.5 ^d^	22.7 ± 3.0 ^d^	34.7 ± 1.3 ^b^	30.7 ± 1.1 ^c^	42.4 ± 0.5 ^a^	36.0 ± 0.4 ^b^	32.2 ± 0.5 ^c^
Glutathione	0.6 ± 0.0 ^d^	0.7 ± 0.0 ^c^	0.8 ± 0.0 ^a^	0.7 ± 00 ^c^	0.8 ± 0.0 ^a^	0.7 ± 0.0 ^b^	0.6 ± 0.0 ^e^
Cystine	0.4 ± 0.0 ^d^	0.4 ± 0.0 ^cd^	0.6 ± 0.0 ^a^	0.4 ± 0.0 ^d^	0.5 ± 0.0 ^b^	0.4 ± 0.0 ^c^	0.4 ± 0.0 ^d^
Total non-EAA	97.4 ± 1.2 ^e^	87.4 ± 9.2 ^f^	137.8 ± 2.9 ^b^	114.5 ± 0.9 ^d^	148.5 ± 1.6 ^a^	121.9 ± 1.0 ^c^	113.2 ± 2.2 ^d^
Total AA	187.4 ± 2.6 ^e^	163.9 ± 18.9 ^f^	246.3 ± 5.9 ^b^	213.5 ± 2.0 ^d^	279.6 ± 3.3 ^a^	227.6 ± 2.1 ^c^	209.9 ± 5.2 ^d^

Values are presented as the mean ± standard deviation of determinations made in triplicate. Significant differences (*p* < 0.05) are indicated by different letters in the same row. ns, non-significant differences. Experimental diet nomenclature: See Table 1.

**Table 7 foods-12-03474-t007:** Recommended amino acid scores for adult (mg/g protein) sago palm weevil larvae fed different diets compared to the reference protein.

Essential Amino Acid	C	PF	F1	F2	F3	F4	F5	Reference Protein ^#^
Valine	41.7 ± 0.2 ^d^ (1.04) *	43.3 ± 5.1 ^cd^ (1.08)	51.2 ± 1.4 ^a^ (1.28)	43.9 ± 1.0 ^cd^ (1.10)	49.5 ± 0.6 ^ab^ (1.24)	46.2 ± 0.7 ^bc^ (1.16)	46.6 ± 1.5 ^bc^ (1.17)	40
Leucine	59.4 ± 0.2 ^d^ (0.97)	60.4 ± 7.7 ^cd^ (0.99)	73.4 ± 2.1 ^a^ (1.20)	61.2 ± 1.2 ^cd^ (1.00)	68.5 ± 1.2 ^ab^ (1.12)	65.7 ± 0.8 ^bc^ (1.08)	64.6 ± 2.2 ^bcd^ (1.06)	61
Isoleucine	33.5 ± 0.9 ^d^ (1.12)	35.1 ± 4.4 ^cd^ (1.17)	42.0 ± 0.9 ^a^ (1.40)	35.3 ± 0.6 ^cd^ (1.18)	39.5 ± 0.4 ^ab^ (1.32)	37.3 ± 0.7 ^bc^ (1.24)	37.5 ± 1.0 ^bc^ (1.25)	30
Threonine	21.6 ± 0.4 ^c^ (0.87)	22.4 ± 2.1 ^c^ (0.90)	26.0 ± 0.6 ^a^ (1.04)	22.0 ± 0.2 ^c^ (0.88)	24.4 ± 0.5 ^b^ (0.98)	23.3 ± 0.5 ^bc^ (0.93)	23.1 ± 0.4 ^bc^ (0.92)	25
Lysine	197.3 ± 4.9 ^c^ (4.11)	194.3 ± 26.2 ^c^ (4.05)	261.5 ± 8.2 ^a^ (5.45)	200.3 ± 5.7 ^c^ (4.17)	246.9 ± 4.9 ^a^ (5.14)	225.0 ± 6.5 ^b^ (4.69)	222.1 ± 7.9 ^b^ (4.63)	48
Histidine	26.8 ± 0.3 ^d^ (1.68)	28.4 ± 3.2 ^cd^ (1.77)	34.1 ± 0.7 ^a^ (2.13)	28.9 ± 0.3 ^cd^ (1.80)	31.7 ± 0.2 ^b^ (1.98)	30.5 ± 0.2 ^bc^ (1.91)	30.2 ± 0.7 ^bc^ (1.89)	16
Tryptophan	0.6 ± 0.0 ^d^ (0.10)	0.8 ± 0.0 ^b^ (0.12)	0.8 ± 0.0 ^a^ (0.12)	0.7 ± 0.0 ^cd^ (0.10)	0.7 ± 0.0 ^b^ (0.11)	0.7 ± 0.0 ^c^ (0.10)	0.7 ± 0.0 ^c^ (0.10)	6.6
Phenylalanine + tyrosine	34.0 ± 0.4 ^d^ (0.83)	35.6 ± 3.4 ^cd^ (0.87)	41.8 ± 0.8 ^a^ (1.02)	34.9 ± 0.7 ^cd^ (0.85)	38.9 ± 0.8 ^b^ (0.95)	37.3 ± 0.8 ^bc^ (0.91)	37.4 ± 1.0 ^bc^ (0.91)	41
Methionine + cysteine	7.0 ± 0.0 ^c^ (0.30)	6.7 ± 0.8 ^c^ (0.29)	8.8 ± 0.2 ^a^ (0.38)	7.0 ± 0.1 ^c^ (0.30)	8.1 ± 0.2 ^b^ (0.35)	7.8 ± 0.1 ^b^ (0.34)	7.6 ± 0.2 ^b^ (0.33)	23
Total EAA	422.0 ± 6.6 ^d^ (1.45)	427.0 ± 52.7 ^d^ (1.47)	539.8 ± 12.7 ^a^ (1.86)	434. ± 5.0 ^cd^ (1.49)	508.1 ± 8.0 ^ab^ (1.75)	473.8 ± 8.5 ^bc^ (1.63)	469.8 ± 14.4 ^bc^ (1.62)	290.6

Values are presented as the mean ± standard deviation of determinations made in triplicate. Significant differences (*p* < 0.05) are indicated by different letters in the same row. Experimental diet nomenclature: See Table 1. * Numbers in parentheses represent the score ratio for each EAA in SPWL fed different diets and in the reference protein (EAA score). ^#^ Food and Agriculture Organization of the United Nations (Rome) [57].

## Data Availability

Data are contained within the article.

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
