# Peer review of "Balancing the Growth Performance and Nutritional Value of Edible Farm-Raised Sago Palm Weevil (Rhynchophorus ferregineus) Larvae by Feeding Various Plant Supplemented-Sago Palm Trunk Diets"

_foods, 2023, doi:10.3390/foods12183474_

Round 1

Reviewer 1 Report

Comments and suggestions for authors

Insect is an ideal food resource for fish industry. The manuscript 'Balancing the Growth Performance and Nutritional Values of Edible Farm-Raised Sago Palm Weevil (Rhynchophorus ferregineus) Larvae by Feeding Various Plant Supplemented-Sago Palm Trunk Diets' discloses the effect of plant-based ingredients on the nutritional values of sago palm weevil larvae. The well-written manuscript focuses on an interesting and topical subject and is of considerable practical importance; however, it is still immature.

LINE 32 What consumers want in terms of nutritional value and can plant-based ingredients solve it?

LINE 32  More data details on the sago palm weevil as an edible insect should be provided in the Introduction. The food safety of SPWL needs to be clarified. For broad readership, it is very important to cite and provide more information about toxicological characteristics of edible insects in Asian and Central and South America because edible insect is not a new product.

1) Feng, Y., Zhao, M., Ding, W.F., & Chen, X.M. (2020). Overview of edible insect resources and common species utilisation in China. Journal of Insects as Food and Feed, 6(1), 13–25. doi: 10.3920/JIFF2019.0022

2) Gao, Y., Wang, D., Xu, M.L., Shi, S.S., & Xiong, J.F. (2018). Toxicological characteristics of edible insects in China: A historical review. Food and Chemical Toxicology, 119, 237–251. doi: 10.1016/j.fct.2018.04.016

3) Yi, C.H.; He, Q.J.; Lin, W.; Kuang, R.P. The utilization of insect-resources in Chinese rural area. Journal of Agricultural Science, 2010, 2, 146–154. https://doi.org/10.5539/jas.v2n3p146

LINE 63-65  Please provide the citation.

LINE 166  Please provide the information on manufacturer of spss.

LINE 696  How much does PI cost and PI contribute to yield improvement?

Author Response

Reviewer#1

Insect is an ideal food resource for fish industry. The manuscript 'Balancing the Growth Performance and Nutritional Values of Edible Farm-Raised Sago Palm Weevil (Rhynchophorus ferregineus) Larvae by Feeding Various Plant Supplemented-Sago Palm Trunk Diets' discloses the effect of plant-based ingredients on the nutritional values of sago palm weevil larvae. The well-written manuscript focuses on an interesting and topical subject and is of considerable practical importance; however, it is still immature.

Ans: Thank you very much for your invaluable suggestion.

LINE 32 What consumers want in terms of nutritional value and can plant-based ingredients solve it?

Ans: The statement was added. “The use of plant-based ingredients (PI), particularly rice bran (RB), soybean meal (SM), and perilla seed (PS), may satisfy the nutritional values that consumers needed from SPWL, such as high protein, rich in essential amino acids/fatty acids, balance in n3/n6 PUFA, low cholesterol, and rich in minerals.

LINE 32  More data details on the sago palm weevil as an edible insect should be provided in the Introduction. The food safety of SPWL needs to be clarified. For broad readership, it is very important to cite and provide more information about toxicological characteristics of edible insects in Asian and Central and South America because edible insect is not a new product.

1) Feng, Y., Zhao, M., Ding, W.F., & Chen, X.M. (2020). Overview of edible insect resources and common species utilisation in China. Journal of Insects as Food and Feed, 6(1), 13–25. doi: 10.3920/JIFF2019.0022

2) Gao, Y., Wang, D., Xu, M.L., Shi, S.S., & Xiong, J.F. (2018). Toxicological characteristics of edible insects in China: A historical review. Food and Chemical Toxicology, 119, 237–251. doi: 10.1016/j.fct.2018.04.016

3) Yi, C.H.; He, Q.J.; Lin, W.; Kuang, R.P. The utilization of insect-resources in Chinese rural area. Journal of Agricultural Science, 2010, 2, 146–154. https://doi.org/10.5539/jas.v2n3p146

Ans: The information was expanded upon using the indicated references as well as additional relevant ones. Additionally, the citations were updated.

LINE 63-65  Please provide the citation.

Ans: Done.

LINE 166  Please provide the information on manufacturer of spss.

Ans: It was changed to “The Statistical Package for the Social Sciences (SPSS) 24 for Windows (SPSS Inc., Chicago, IL, USA) was used for statistical analysis.”

LINE 696  How much does PI cost and PI contribute to yield improvement?

Ans: Each ingredient's cost was listed in the Materials and Methods. “The costs of GSPT, PF, RB, SM, and PS were approximately 0.13, 0.22, 0.26, 0.32, and 1.70 US$/kg [15].

For the yield, the statement was added in the Conclusion. “In comparison to the commercial PF diet and the typical GSPT diet, PI increased the yield expressed as live weight by up to two folds.

Reviewer 2 Report

Regarding the manuscript entitled Balancing the Growth Performance and Nutritional Values of Edible Farm-Raised Sago Palm Weevil (Rhynchophorus ferregineus) Larvae by Feeding Various Plant Supplemented-Sago Palm Trunk Diets

L28. Add conclusion at the end of abstract.

L73. Add hypothesis

In the introduction section, the authors should highlight the novelty of their study and why they chose these feed ingredients.

L85. Why pig feed is considered a positive control, it should be a treatment group to compare with.

L163. Add more details.

Table 1 should be after its first mention, in the material and methods section please move it. Or separate it into two tables, feed ingredients in one table and composition in another.

The discussion section is weak and needs more work from the authors, The Results and Discussion section mainly describes the study's findings, ignoring the discussion. This section has very long paragraphs, please try to improve this section.

L266. This section and others are very long, please divide them into two paragraphs.

moderate editing

Author Response

Reviewer#2

Regarding the manuscript entitled Balancing the Growth Performance and Nutritional Values of Edible Farm-Raised Sago Palm Weevil (Rhynchophorus ferregineus) Larvae by Feeding Various Plant Supplemented-Sago Palm Trunk Diets

L28. Add conclusion at the end of abstract.

Ans: At the end of the abstract, the conclusion was added. “Therefore, incorporating PI into a regular diet is a viable method for enhancing the nutritional value and sustainability of farm-raised SPWL as a potential alternative source of high-quality lipid and protein.

L73. Add hypothesis

Ans: The hypothesis was added. “It was predicated on the hypothesis that dietary intake significantly affects how well SPWL grows and how nutritious it is.

In the introduction section, the authors should highlight the novelty of their study and why they chose these feed ingredients.

Ans: The research challenge was posed in the thrid paragraph of the Introduction based on the prior reports indicating diet has a substantial impact on how well SPWL grows and how nutritious it is. The benefits and drawbacks of feeding SPWL with rice bran, soybean meal, and perilla seed were discussed. This can lead to a solution that makes use of a diversified diet to maximize nutrients and growth prospects, as was noted in the last section of this paragraph before establishing the goal in the next one.

L85. Why pig feed is considered a positive control, it should be a treatment group to compare with.

Ans: In order to make this point more clearly the explanation was altered to “The optimization of supplementing plant-based ingredients (PI) in the basal ground sago palm trunk (GSPT; 66.7%) with varying concentrations of soybean meal (SM; 4.4-11.1%), rice bran (RB; 8.9-22.2%), and perilla seed (PS; 0-20%) was investigated (Table 1) in comparison to the basal diet alone (Control; C), and the basal diet supplemented with pig feed (PF). Two types of feed are often utilized on farms to feed SPWL, depending on the growing region: (1) GSPT alone, which represented natural raising, and (2) GSPT enriched with PF, which is frequently added to the basal diet in many farm-raised SPWL to boost growth rate. One of the factors that affects their utilization is religious conviction. For instance, farmers who are not restricted by their religious beliefs supplemented PF to their feeds, whereas Muslim farmers are prohibited from doing so. Thus, both diets were included in this study. The nutritional content, growth performance, lipid metabolism indicator, and gene expression of D6 desaturase (fads 2) in SPWL were all thoroughly examined. To prepare supplemental diets, GSPT was mixed with each supplement in the specific content shown in Table 1 until the feed was homogeneous. PS was purchased from Bankongloi, Chiangmai, Thailand, and PF, RB, and SM were purchased from Thasala market, Nakhon Si Thammarat, Thailand.”

L163. Add more details.

Ans: The Fads2 Gene Expression of SPWL was described in detail as recommended.

Table 1 should be after its first mention, in the material and methods section please move it. Or separate it into two tables, feed ingredients in one table and composition in another.

Ans: Table 1 was separated into 2 Tables.

The discussion section is weak and needs more work from the authors, The Results and Discussion section mainly describes the study's findings, ignoring the discussion. This section has very long paragraphs, please try to improve this section.

Ans: The invention of a supplementary plant-based diet for this type of insect was initially reported in this article. In order to declare the discovery and allow the simplifier to be used realistically, the result was thoroughly described. The discussion was also centered on the connections to earlier works that had been published as well as the potential mechanisms by which those substances might affect the rate of growth and nutritional value of this insect. 

L266. This section and others are very long, please divide them into two paragraphs.

Ans: Done.

Reviewer 3 Report

The manuscript titled “Balancing the Growth Performance and Nutritional Values of Edible Farm-Raised Sago Palm Weevil (Rhynchophorus ferregineus) Larvae by Feeding Various Plant Supplemented-Sago Palm Trunk Diets” was aimed to fine-tune the proper feed composition using agricultural plant-based ingredients (soybean meal, rice bran, and perilla seed) to balance the growth performance and nutritional values of SPWL while avoiding any negative effects. The topic is relevant and the manuscript deserves consideration. The work is well designed and the results obtained are interesting. However, at presented state the manuscript needs revision. The main comments and recommendations are listed below.

L. 41-43. “…(SPWL) are widely consumed in numerous regions of the world”. It is better to mention regions that already cultivate and consume SPWL. This insect has a high nutritional value and has a potential to be consumed in all over the World. However, at the moment SPWL growth and are used as food or feed only in several regions of the World. They are still not so common used as BSFL, grasshoppers, locusts, mealworms etc.    

Introduction should be modified: expanded and reached by recent and relevant references. Paragraph 2 has to less references. Perhaps, it is associated with small number of articles regarding SPW and similar insects larvae as food or feed. If so, the authors missed recent review of Rhynchophorus ferrugineus (Olivier) (Coleoptera: Curculionidae) as human food – it can be considered in the text (https://doi.org/10.3920/JIFF2022.0095).

What about consumer acceptability? The authors mentioned it in Results and Discussion section, but it was is related to usage of commercial pig feed formulated diets… The authors should discuss consumer acceptance of edible insects and SPWL, in particular, in Introduction. For example, https://doi.org/10.1007/s42690-021-00487-7

Some review and discussion on successful manipulation of nutritional value of edible insect larvae can be added as separate paragraph to make a logical passing to the last paragraph. 

L. 99. Technological parameters for blanching and freeze drying should be given. What equipment the authors used?

L. 117. “Minerals were identified using inductive couple plasma (ICP) spectrometry [7].” After checking Reference list, I understood that it is reference to the method, but not to the equipment details. It is better to add “…using method [7]”. Additionally, the authors should add details for ICPS: model/brand (Manufacturer, City, Country).

The authors should provide such details for all equipment used in the experiment. This is very important to assess reproducibility of the experiment. For reproducibility of the experiment the authors should not miss parameters of measurement and details for chemicals (chemical grade, Manufacturer, City, Country). This should be checked in the whole text and corrected.

Results and Discussion section is well structured and presented. All data obtained are statistically processed to justify significance of the results. As recommendation, the authors can reach Discussion part with comparison of results obtained towards nutritional compositions of SPWL with results of other researchers.

L. 261. “Growth performance of sago palm weevil larvae (SPWL) fed with different diets compared to referent protein”. The authors can remove abbreviation from the title since they did not use it in the table 2. The same for tables 3-6 and figure 1.

Conclusions should be supported by data as the authors well reached the Abstract by data.

The manuscript should be checked for insignificant typos correction.

Generally, the work is interesting and the results presented are interesting. The manuscript can be reconsidered after revision.

The manuscript should be checked for insignificant typos correction.

Author Response

Reviewer#3

The manuscript titled “Balancing the Growth Performance and Nutritional Values of Edible Farm-Raised Sago Palm Weevil (Rhynchophorus ferregineus) Larvae by Feeding Various Plant Supplemented-Sago Palm Trunk Diets” was aimed to fine-tune the proper feed composition using agricultural plant-based ingredients (soybean meal, rice bran, and perilla seed) to balance the growth performance and nutritional values of SPWL while avoiding any negative effects. The topic is relevant and the manuscript deserves consideration. The work is well designed and the results obtained are interesting. However, at presented state the manuscript needs revision. The main comments and recommendations are listed below.

Ans: We appreciate your thoughtful advice on how to make our manuscript better.

  1. 41-43. “…(SPWL) are widely consumed in numerous regions of the world”. It is better to mention regions that already cultivate and consume SPWL. This insect has a high nutritional value and has a potential to be consumed in all over the World. However, at the moment SPWL growth and are used as food or feed only in several regions of the World. They are still not so common used as BSFL, grasshoppers, locusts, mealworms etc.

Ans: It was changed to “Because of their excellent nutritional content and distinct taste and flavor, SPWL are widely consumed in numerous regions of the world, particularly in Asia and Africa [8, 9, 10, 11]. The trade in edible insects, specifically SPWL, holds immense promise because some countries, like Thailand, even grow SPWL on a commercial scale for the food industry [12].” The references were also updated.

Introduction should be modified: expanded and reached by recent and relevant references. Paragraph 2 has to less references. Perhaps, it is associated with small number of articles regarding SPW and similar insects larvae as food or feed. If so, the authors missed recent review of Rhynchophorus ferrugineus (Olivier) (Coleoptera: Curculionidae) as human food – it can be considered in the text (https://doi.org/10.3920/JIFF2022.0095).

Ans: As advised, the Introduction was enhanced. To assure the prospective use of this insect with food safety concerns as indicated by reviewer#1, the proposed review article and other relevant references were incorporated.

What about consumer acceptability? The authors mentioned it in Results and Discussion section, but it was is related to usage of commercial pig feed formulated diets… The authors should discuss consumer acceptance of edible insects and SPWL, in particular, in Introduction. For example, https://doi.org/10.1007/s42690-021-00487-7

Ans: The Introduction was expanded to provide information about how well-liked consumers find products created with SPWL and related African palm weevil larvae.

Some review and discussion on successful manipulation of nutritional value of edible insect larvae can be added as separate paragraph to make a logical passing to the last paragraph.

Ans: Done.

  1. 99. Technological parameters for blanching and freeze drying should be given. What equipment the authors used?

Ans: It was changed to “To determine nutritional values, 40-day-old larvae were rinsed with tap water, blanched (boiling water/5 min), cooled down (ice water/10 min), freeze-dried using a freeze-dryer (FTS systems Inc., Stone Ridge, NY, USA), and kept in a plastic box at -20 °C until used.”

  1. 117. “Minerals were identified using inductive couple plasma (ICP) spectrometry [7].” After checking Reference list, I understood that it is reference to the method, but not to the equipment details. It is better to add “…using method [7]”. Additionally, the authors should add details for ICPS: model/brand (Manufacturer, City, Country).

Ans: It was changed to “Minerals were identified by the AOAC method [17], using an inductively coupled plasma optical emission spectrophotometer (PerkinElmer, Model 4300DV, Norwalk, CT, USA).

The authors should provide such details for all equipment used in the experiment. This is very important to assess reproducibility of the experiment. For reproducibility of the experiment the authors should not miss parameters of measurement and details for chemicals (chemical grade, Manufacturer, City, Country). This should be checked in the whole text and corrected.

Ans: It was stated in Section 2.1 that “The solvents and chemicals used in this investigation were all gas chromatography (GC) and analytical grade, purchased from Sigma-Aldrich (St. Louis, MO, USA).

Furthermore, all of the equipment used in the experiment was described in detail throughout the text.

Results and Discussion section is well structured and presented. All data obtained are statistically processed to justify significance of the results. As recommendation, the authors can reach Discussion part with comparison of results obtained towards nutritional compositions of SPWL with results of other researchers.

Ans: Thank you very much. Since there was not much data on the impact of diet composition on growth performance and nutritional value of SPWL. As a result, the discussion was related to other insect species and evaluated against our earlier published data. Discussion topics encompassed both the proposed mechanism and the basic scientific idea.

  1. 261. “Growth performance of sago palm weevil larvae (SPWL) fed with different diets compared to referent protein”. The authors can remove abbreviation from the title since they did not use it in the table 2. The same for tables 3-6 and figure 1.

Ans: As suggested the abbreviation was taken out of the titles of the Tables and Figure. 

Conclusions should be supported by data as the authors well reached the Abstract by data.

Ans: The data were used to support the conclusion, which was concise and informative. Additionally, the significance of the findings was highlighted. “The diet composition is critical in determining the growth performance and nutritional values of SPWL. The balance of RB, SM, and PS with GSPT diet resulted in improved protein and lipid quantities and qualities of SPWL, as well as increased macro- and micro-minerals which stimulated growth and had a higher survival rate than the commercial PF diet and the regular GSPT diet. In comparison to the commercial PF diet and the typical GSPT diet, PI increased the yield expressed as live weight by up to two folds. Furthermore, optimizing feed composition can lead to increased production efficiency and sustainability of farm-raised SPWL production. Overall, incorporating PI into a regular diet is a promising strategy for improving the nutritional quality and sustainability of farm-raised SPWL as a potential alternative protein and lipid source.

The manuscript should be checked for insignificant typos correction.

Ans: To make the manuscript consistent and comprehensible, all of the typos were examined again.

Generally, the work is interesting and the results presented are interesting. The manuscript can be reconsidered after revision.

Ans: Thank you very much.

Round 2

Reviewer 2 Report

Thank you for your revisions 

Minor editing 

Reviewer 3 Report

The revised manuscript can be recommended for consideration for publication in Foods